# Revisiting Padded Transformer Expressivity:
# Which Architectural Choices Matter and Which Don't

**Anej Svete** [1]  **William Merrill** [2]  **Ryan Cotterell** [1]  **Ashish Sabharwal** [2]

## Abstract

Recent work describes what transformers can and cannot compute through connections to boolean circuits, but existing results lack exact characterizations and are sensitive to modeling choices. *Padded* transformers—to whose input filler symbols such as "..." are appended—emerge as a useful gadget for establishing *equivalences* to circuit classes by providing polynomial space for adaptive parallel computation. However, only a limited set of padded transformer idealizations has been studied, leaving open how robustly these equivalences hold under changes to attention type, model width, and uniformity. We find that, under practical assumptions, padded transformers are surprisingly *robust* to all of these, and identify numeric precision and model depth as the main factors affecting expressivity. Concretely, we prove that polynomially padded L-uniform constant-precision transformers are equivalent to L-uniform $AC^0$, while growing-precision ones achieve L-uniform $TC^0$ regardless of width. Furthermore, *looping* enables sequential processing analogous to circuits: $\log^d N$-looped constant-precision transformers reach FO-uniform $AC^d$, and growing-precision ones reach FO-uniform $TC^d$. Interestingly, growing width or precision beyond logarithmic does not increase expressivity, and all our results hold for *both* softmax and average hard attention transformers.

## 1. Introduction

A large body of work explores transformer expressivity, i.e., what functions transformers can and cannot compute. Because such work is formal, one must mathematically pin down all aspects of the model. However, it has become apparent that transformer expressivity can be highly sensitive to some design choices; for example, the type of attention—soft versus hard—has severe consequences on what languages transformers can recognize (Hao et al., 2022; Jerad et al., 2025). This has led to diverse and sometimes difficult-to-reconcile results on transformer expressivity, where seemingly similar transformer variants—differing only slightly in, say, numeric precision assumptions—achieve substantially different expressivity (Strobl et al., 2024).

In stark contrast to this brittleness, we show that *padded transformers* (Pfau et al., 2024)—which append polynomially many dedicated filler symbols □ to the input before processing it—are surprisingly *robust* to a variety of changes to model specification, including attention type, model width, and parameter uniformity. Numeric precision and model depth, in contrast, emerge as major factors determining expressivity; log-precision padded transformers are always more expressive than constant-precision ones, and expressivity grows with model depth. Once logarithmic precision is achieved in an L-uniform padded transformer (a transformer that can be constructed by a logspace Turing machine; cf. Def. 2.4), however, other aspects like attention type, model width, or further increase in precision do not affect expressivity. This may be attractive to both theorists and practitioners: On the one hand, it simplifies theoretical analyses, which can focus on whichever equivalent specification is easiest to study; on the other hand, it suggests that the derived characterizations are more likely to describe real-world models.

Intuitively, padding abstracts ways of adaptively increasing parallel inference-time computation, such as pause symbols (Pfau et al., 2024) and text diffusion models (Svete & Sabharwal, 2026), which improve the empirical performance of transformers on a variety of tasks (Pfau et al., 2024; Goyal et al., 2024; London & Kanade, 2025). It has also proven to be a useful theoretical gadget for studying transformer expressivity, yielding *exact* expressivity characterizations—but only for specific choices of uniformity, precision, width, and attention type (Li et al., 2024b; Merrill & Sabharwal, 2025a; London & Kanade, 2025), leaving open how robust these characterizations are. Our comprehensive analysis of a large set of possible transformer idealizations (cf. Fig. 1)

[1]ETH Zürich [2]Allen Institute for AI. Correspondence to: Anej Svete <asvete@ethz.ch>, Ryan Cotterell <ryan.cotterell@ethz.ch>.

*Proceedings of the 43rd International Conference on Machine Learning*, Seoul, South Korea. PMLR 306, 2026. Copyright 2026 by the author(s).

| Depth | Precision ($b$) | Fully uniform AHATs $\Theta(1)$ | Width ($D$) — L-uniform AHATs *and* SMATs | | |
|---|---|---|---|---|---|
| | | | $\Theta(1)$ | $\Theta(\log N)$ | $\texttt{poly}(N)$ |
| $\Theta(1)$ | $\Theta(1)$ | | | $\star$ | L-uniform $\texttt{AC}^0$ — AC/TC divide |
| | $\Theta(\log N)$ | † FO-uniform $\texttt{TC}^0$ | | $\star$ | L-uniform $\texttt{TC}^0$ |
| | $\texttt{poly}(N)$ | † | | | |
| $\Theta(\log^d N)$ $(d \geqslant 1)$ | $\Theta(1)$ | | | FO-uniform $\texttt{AC}^d$ — AC/TC divide | |
| | $\Theta(\log N)$ | † | FO-uniform $\texttt{TC}^d$ | | |
| | $\texttt{poly}(N)$ | † | | | |

*Figure 1.* Polynomially padded transformer expressivity across depth ($\downarrow$), precision ($\downarrow$), uniformity ($\rightarrow$), and width ($\rightarrow$) regimes, ignoring parameterizations that do not satisfy the sufficient volume constraint (cf. Def. 2.3). In contrast to most existing results on transformer expressivity, these results are *exact* for padded transformers. The purple lines mark the AC/TC *divide*: Constant-precision transformers are limited to $\texttt{AC}^d$, while growing-precision transformers achieve $\texttt{TC}^d$. † marks Merrill & Sabharwal's (2025a) results on fully uniform AHATs (average hard attention transformers) and $\star$ London & Kanade's (2025) results on L-uniform SMATs (soft attention transformers).

reveals the previously unknown robustness of padded transformers to these differences.

Padding facilitates a particularly convenient connection to **boolean circuits**—computational models that process fixed-length inputs through layers of logic gates in the form of an acyclic graph (Hao et al., 2022; Merrill & Sabharwal, 2023; Li et al., 2024b; London & Kanade, 2025, *inter alia*). Natural and well-understood examples of circuit classes are $\texttt{AC}^d$—circuits with AND, OR, and NOT gates whose number of gates scales polynomially with string length $N$ and depth with $\log^d N$—and $\texttt{TC}^d$, which add threshold gates that test whether the number of active inputs exceeds some threshold. While connections to circuits have been a fruitful avenue toward understanding transformers, establishing *equivalences* to natural circuit classes is difficult: The attention mechanism's famously *quadratic* complexity in $N$ fundamentally *caps* the parallel computation a transformer can perform at a quadratic amount in $N$. It is unclear how a transformer could execute a cubic or higher-degree polynomial number of parallel computations, making equivalences to natural circuit classes unlikely. Most transformer-to-circuit connections, thus, take the form of (loose) *upper bounds*—showing that transformers can be simulated by either $\texttt{AC}^0$ or $\texttt{TC}^0$ circuits—without matching lower bounds.

We study how prominent aspects of the transformer architecture affect the exact expressivity of polynomially padded transformers to establish a set of equivalences robust to changes in model specification. We focus on *(1)* the type of attention (softmax attention (SMAT) and average hard attention (AHAT) transformers), *(2)* the scaling of numeric precision, width, and depth, and *(3)* the uniformity of the transformer constructions. We find order in existing literature by shifting focus to padded uniform transformer families. Studying *families* is necessary since letting transformer parameters depend on context length requires constructing a

separate model for each length.[1] A uniform family describes how each model is built. We study L-uniform families (London & Kanade, 2025), which enforce that the transformers can be built by a simple computational model—a logspace Turing machine. We build on London & Kanade's (2025) characterizations of padded L-uniform SMATs and Merrill & Sabharwal's (2025a) padded log-precision *fully* uniform AHATs (where one set of parameters works for every input length) (cf. †- and $\star$-marked cells in Fig. 1). By connecting AHATs and SMATs, translating London & Kanade's (2025) results to AHATs, and extending Merrill & Sabharwal's (2025a) results to constant-precision transformers, we establish the following results, summarized in Fig. 1:

(1) **Constant vs. growing precision** and the AC/TC expressivity divide: A consistent trend in the effect of numeric precision appears: It determines whether the equivalence is to L-uniform $\texttt{AC}^0$ (with constant precision) or to L-uniform $\texttt{TC}^0$ circuits (with growing precision).

(2) We find **robustness with padding**: A particularly important quantity turns out to be the transformer's **volume** $V(N) \overset{\text{def}}{=} b \cdot D$—the number of bits available at each symbol at each layer of the transformer—where $b$ denotes the numeric precision and $D$ the model width. As long as the volume is at least $\Omega(\log N)$ (which is required to distinguish $N$ input positions), L-uniform padded transformers are *robust* to changes in their specification: They either match L-uniform $\texttt{AC}^0$ (with constant precision) or L-uniform $\texttt{TC}^0$ (with growing precision). The type of attention, width, and

---

[1]Without uniformity constraints, transformers and circuits are unrealistically powerful. For example, consider the unary language $\{1^N \mid$ the $N^{\text{th}}$ Turing machine halts$\}$ under some fixed enumeration of Turing machines. This undecidable language is recognizable by a non-uniform circuit family, since we can hard-code the correct answer for each input length $N$ into the circuit $C_N$. Uniformity conditions prevent such pathological cases by requiring a single, feasible algorithm to construct all circuits in the family.

precision beyond logarithmic *do not affect* expressivity.

(3) **Natural scaling with looping**: *Looping* endows transformers with both parallel and sequential processing, enabling them to recognize regular (Merrill & Sabharwal, 2025b) and context-free (Jerad et al., 2026) languages that constant-depth transformers cannot. We extend the characterizations of constant-depth transformers to looped ones, showing that their expressivity scales analogously to circuits under all regimes: $\Theta(\log^d N)$-looped constant-precision transformers achieve FO-uniform $\mathrm{AC^d}$ while growing-precision ones achieve FO-uniform $\mathrm{TC^d}$. As $d \to \infty$ both approach $\mathrm{NC} = \bigcup_{d \geqslant 0} \mathrm{AC^d} = \bigcup_{d \geqslant 0} \mathrm{TC}^d$.

(4) With growing precision, there is **no benefit to growing width or weakening uniformity**: Fully uniform and L-uniform growing-precision padded transformers with $\Theta(\log^d N)$ looping are equivalent to FO-uniform $\mathrm{TC^d}$ regardless of the width. There is also no benefit to polynomial precision over logarithmic precision.

(5) Describing the expressivity of transformers with **insufficient ($o(\log N)$) volume** is difficult with natural circuit classes. Understanding sub-classes of $\mathrm{AC^d}$ is likely required.

## 2. Preliminaries

Here, we introduce the core objects of the paper: The two attention variants (SMAT and AHAT), the width, precision, and volume of a transformer, uniform transformer families, fixed-point arithmetic, and looped padded transformers. We follow Merrill & Sabharwal (2025a) and London & Kanade (2025) in our formal exposition. We outline the setup here and give additional details in §A. We study softmax attention transformers (SMATs), which are commonly used in practice, and average hard attention transformers (AHATs), which are often preferred in the theoretical literature. Both can be defined via **temperature-scaled softmax attention**, which computes the attention weights for input of length $N \in \mathbb{N}$ based on unnormalized attention scores $\boldsymbol{x} \in \mathbb{R}^N$, position $n \in [N]$, and **temperature** $\tau > 0$ as

$$\mathrm{softmax}_\tau(\boldsymbol{x})_n = \frac{\exp(x_n/\tau)}{\sum_{n'=1}^N \exp(x_{n'}/\tau)}. \tag{1}$$

We treat the temperature $\tau = \tau(N)$ as a model parameter that may depend on the input length, analogously to the parameter matrices and PEs of $\mathcal{T}_N$; concretely, we require $\tau(N)$ to be computable in the same complexity class as the rest of the model (cf. Def. A.5). This treatment of the temperature as a computable parameter mirrors the role that it plays in practical attention implementations, and captures the standard way of approximating AHATs with SMATs (Yang et al., 2026a)—the $\tau \to 0$ limit of SMATs yields AHATs:

$$\lim_{\tau \to 0} \mathrm{softmax}_\tau(\boldsymbol{x})_n = \begin{cases} \frac{1}{|\operatorname{argmax}(\boldsymbol{x})|} & \textbf{if } x_n = \max(\boldsymbol{x}), \\ 0 & \textbf{otherwise}. \end{cases} \tag{2}$$

One of the main aims of the paper is to unify the literature on SMATs and AHATs. Throughout the paper, all statements referencing transformers hold for both attention types; we only specify the type when the distinction is necessary.

We use $D$ to denote the model width, i.e., the size of each symbol's internal representation (the residual stream), and $b$ to denote the numeric precision of the model's computations, i.e., the number of bits with which each scalar in the model is stored and manipulated (cf. §A.3).

**Definition 2.1** (Width and precision). *The **width** $D(N) \in \mathbb{N}$ of a transformer is the dimensionality of each symbol's residual-stream representation. The **numeric precision** $b(N) \in \mathbb{N}$ is the number of bits used to store and manipulate every scalar in the model under fixed-point arithmetic (cf. §A.3). Both may depend on the input length $N$.*

As standard in theoretical literature (e.g., Li et al., 2024b), we allow $D$ and $b$ to *grow* with input length $N$. This is required even just to *name* the $N$ input positions: Distinguishing $N$ positions requires $\log N$ bits per symbol, so a transformer must satisfy $D(N) \cdot b(N) = \Omega(\log N)$ to encode uncompressed positional information. This motivates the definition of the transformer (representation) volume.

**Definition 2.2** (Volume). *The **volume** of a transformer is*

$$V(N) \stackrel{\text{def}}{=} D(N) \cdot b(N). \tag{3}$$

Intuitively, this corresponds to the logarithm of the number of distinct attention query vectors (whose $D$ entries are stored as a $b$-bit number) at any position in a string of length $N$. One of our main results reveals that, as long as a polynomially padded transformer has volume $V(N) = \Omega(\log N)$, additional model growth does not increase expressivity. We capture this in the following definition.

**Definition 2.3** (Sufficient volume). *A transformer family has **sufficient volume** if its volume satisfies $V(N) = \Omega(\log N)$, and **insufficient volume** otherwise.*

What matters, though, is where the growing volume comes from—growing-precision transformers can be more expressive than growing-width ones.

**Transformer families.** Making the width a function of the input length means that each input length is processed by a *separate* transformer $\mathcal{T}_N$, yielding a **family** $\{\mathcal{T}_N\}_{N \in \mathbb{N}}$ of transformers. In a general transformer family, there need not be any relationship between the transformers of different sizes, resulting in similar pathologies as in non-uniform circuit classes (cf. Footnote 1). Thus, we enforce L-**uniformity** in our constructions, which requires that a logspace Turing machine can construct $\mathcal{T}_N$ from input $1^N$ (cf. §A.4). This means that all transformers are easily algorithmically constructible. We contrast L-uniform transformer families to **fully uniform** families, in which a *single* transformer $\mathcal{T}$

(with one fixed set of parameters and a single positional-encoding scheme) processes inputs of every length $N \in \mathbb{N}$. In the language of §A.4, this means that the Turing machine computing the transformer parameters is a *constant function* in $N$. Fully uniform families are thus more restrictive than L-uniform ones, since the latter allow the description of $\mathcal{T}_N$ to depend on $N$ (via a logspace Turing machine), whereas the former require a length-independent description.

**Definition 2.4** (Uniform transformer families; variant of London & Kanade, 2025, Def. 3.6)**.** *A transformer family* $\{\mathcal{T}_N\}_{N \in \mathbb{N}}$ *is* L-**uniform** *if there exist logspace Turing machines* $\mathcal{M}_1$ *and* $\mathcal{M}_2$ *such that:*

*(1)* $\mathcal{M}_1$ *outputs a description of* $\mathcal{T}_N$ *on input* $1^N$, *and*
*(2)* $\mathcal{M}_2$ *outputs* $\boldsymbol{p}(n, N)$ *on input* $(1^N, \mathtt{B}(n))$

*where* $\boldsymbol{p}(n, N)$ *is the positional encoding (PE) at position* $n$ *and* $\mathtt{B}(n) \in \{0, 1\}^{\lceil \log N \rceil}$ *is the binary encoding of* $n$.

**Fixed-point arithmetic.** We model the transformer's computations with fixed-point arithmetic of growing precision. This specifies how model parameters and activations are stored and manipulated. The transformer $\mathcal{T}_N$ with precision $b$ operates over the set $\mathbb{F}_b \stackrel{\text{def}}{=} \{x_\pm \cdot a \cdot 2^{-b} \mid x_\pm \in \{-1, 1\}, a \in \{0, 1, \ldots, 2^{2b} - 1\}\}$—signed fixed-point numbers with $b$ bits each for the integer and fractional parts (cf. §A.3), following the convention of Li et al. (2024b) and London & Kanade (2025). Fixed-point arithmetic both limits the precision of operations and enables the implementation of useful attention gadgets; in particular, the *non-associativity* of fixed-point arithmetic operations enables the implementation of non-linear functions in otherwise linear parts of the model; see §A.3 for details.

**Looped padded transformers (LPTs). Looped** transformers repeatedly apply a fixed block of transformer layers to the input (Dehghani et al., 2019). This dynamically increases model depth, enabling more complex reasoning, while keeping model size constant, thus reducing the memory footprint (Bae et al., 2026). **Padded** transformers additionally pad the input with blank symbols, which can be used to perform additional computations *in parallel*. This additional padding space is analogous to increasing the circuit width in circuit complexity. Concretely, a **looped padded transformer** (LPT) is a triple $(\mathcal{T}, r, P)$ in which $\mathcal{T}$ is a transformer with a designated block of layers, $r \colon \mathbb{N} \to \mathbb{N}$ is the number of times that block is applied (the **loop count**), and $P \colon \mathbb{N} \to \mathbb{N}$ is the **padding length**: On input $\boldsymbol{w} \in \Sigma^N$, the model runs $\mathcal{T}$ on $\boldsymbol{w} \circ \underbrace{\Box \cdots \Box}_{P(N)}$ and applies the looped block of layers $r(N)$ times. See §A.5 for details.

**Boolean circuits** provide a useful abstraction of transformers. They process binary strings by passing them through a directed acyclic graph of nodes that represent boolean operations. Particularly interesting are (1) $\mathsf{AC}^\mathsf{d}$ circuits (Li

et al., 2024b)—boolean circuits of depth $\mathcal{O}(\log^d N)$ ($N$ being the input length), size $\mathtt{poly}(N)$, and with AND, OR, and NOT gates of unbounded fan-in; and (2) $\mathsf{TC}^\mathsf{d}$ circuits, the class of *threshold circuits* of depth $\mathcal{O}(\log^d N)$ and size $\mathtt{poly}(N)$ that additionally allow threshold gates (which determine whether the number of inputs exceeds some threshold). §§ A.2 and A.4 provide more details.

**Notation.** We focus on *polylogarithmically looped* and *polynomially padded* transformers.[2] We use $\mathsf{LPT}^\mathsf{d}_{b,D}$ to refer to transformers with polynomial padding, $\Theta(\log^d N)$ looping, numeric precision $b$, and width $D$. Rather than writing $\Theta(1)$, $\Theta(\log N)$, or $\mathtt{poly}(N)$ for numeric precision and width, we use the shorthand c (constant) for $\Theta(1)$, l (log) for $\Theta(\log N)$, and p (polynomial) for $\mathtt{poly}(N)$, respectively. Thus, e.g., $\mathsf{LPT}^\mathsf{d}_{1,c}$ refers to $\log^d N$-looped log-precision constant-width LPTs with polynomial padding.

## 3. SMATs Can Simulate AHATs

Early work on transformer expressivity focused largely on AHATs (Hao et al., 2022; Merrill et al., 2022; Svete et al., 2024) since their sparser activations make them easier to connect to boolean circuits. Recently, SMATs under different precision regimes have received more attention, providing a new perspective on the expressivity of practical transformers (Merrill & Sabharwal, 2023; Li et al., 2024b; London & Kanade, 2025; Li & Cotterell, 2025; Svete & Sabharwal, 2026). In specific cases, the relationship between the two attention variants is known; for example, Li & Cotterell (2025) show their equivalence in the constant-precision constant-width unpadded regime. Yang et al. (2026a) study *approximating* real-valued AHATs with real-valued SMATs, showing that, by scaling the *temperature* of the softmax normalization in SMATs, one can approximate AHATs arbitrarily well. However, beyond the constant-precision constant-depth regime, no *exact* equivalence results are known.

We bring the two frameworks closer together by showing that their expressivity coincides under logarithmic-precision fixed-point arithmetic. We begin by showing that any AHAT can be simulated by a sufficiently precise SMAT *exactly*.[3]

**Lemma 3.1.** *Let* $\{\mathcal{T}\}_{N \in \mathbb{N}}$ *be a logarithmic-precision* L-*uniform* AHAT *family. There exists a logarithmic-precision* L-*uniform* SMAT *family* $\{\mathcal{T}'\}_{N \in \mathbb{N}}$ *such that for any input* $\boldsymbol{w} \in \Sigma^N$ *and* $N \in \mathbb{N}$, *the outputs of* $\mathcal{T}_N$ *and* $\mathcal{T}'_N$ *match.*

See §C.1 for the full proof. Thus:

**Corollary 3.1.** *For any* $d \in \mathbb{N}$, *we have* L-*uniform* SMAT $\mathsf{LPT}^\mathsf{d}_{*,\dagger} \supseteq$ L-*uniform* AHAT $\mathsf{LPT}^\mathsf{d}_{*,\dagger}$, *where* $*$ *is a placeholder for* l *or* p, $*$ *is a placeholder for*

---

[2]In the following, we will use the term padded transformer to specifically refer to *polynomially* padded transformers.

[3]The proofs of all claims in the main text appear in §C.

c, l, *or* p, *and* SMAT/AHAT *refer to the type of attention in the transformer family.*

Lem. 3.1 lets us translate any (fully uniform or L-uniform) AHAT into an L-uniform SMAT by formalizing the intuition that L-uniform SMATs are at least as powerful as fully-uniform AHATs. This lets us transfer existing expressivity lower bounds on fully-uniform AHATs to L-uniform SMATs as well. It remains unclear whether *fully uniform* SMATs can simulate (fully uniform) AHATs; some degree of non-uniformity seems necessary to implement the temperature scaling required to focus on individual positions.

# 4. Padded Constant-depth Transformers Are Constant-depth Circuits

We now describe the expressivity of constant-depth transformers. Their expressivity under different idealizations has been studied extensively. It is known that fully-uniform log-precision transformers fall under L-uniform $TC^0$ (Merrill & Sabharwal, 2023). Recent work has refined these results, showing that polynomially padded (L-uniform) constant-precision log-width SMATs are equivalent to (L-uniform) $AC^0$, while polynomially padded (L-uniform) log-precision log-width SMATs are equivalent to (L-uniform) $TC^0$ (Li et al., 2024b; London & Kanade, 2025). We restate these results here for reference (see also Fig. 1).

**Theorem 4.1** (London & Kanade, 2025, Thms. 4.1 and 4.5). L-*uniform* $LPT^0_{c,1} =$ L-*uniform* $AC^0$ *and* L-*uniform* $LPT^0_{1,1} =$ L-*uniform* $TC^0$.

Missing, however, is the placement of L-uniform AHATs and *constant-width* SMATs. Here, we extend the known relationships by showing:

• **A lower bound** (§4.1): We tighten the $TC^0$ direction of Thm. 4.1 (i.e., L-uniform $TC^0 \subseteq$ L-uniform $LPT^0_{1,1}$) by showing that *constant*-width log-precision L-uniform transformers already suffice: L-uniform $TC^0 \subseteq$ L-uniform $LPT^0_{1,c}$. This shows that precision can compensate for width. In contrast to Merrill & Sabharwal (2025a), who establish the equivalence of fully-uniform growing-precision AHATs to $F0$-uniform $TC^0$, we connect L-uniform transformers to L-uniform $TC^0$.
• **An upper bound** (§4.2): We show that even with maximal resources—polynomial width in the constant-precision case and polynomial width and polynomial precision in the growing-precision case—transformers cannot exceed L-uniform $AC^0$ or L-uniform $TC^0$, respectively: We have L-uniform $LPT^0_{c,p} \subseteq$ L-uniform $AC^0$ in the constant-precision case and L-uniform $LPT^0_{p,p} \subseteq$ L-uniform $TC^0$ in the growing-precision case.

We show these bounds for *both* AHATs and SMATs, revealing that the distinction between the two is not important in this regime. Together, these bounds sandwich all intermediate regimes, characterizing their expressivity.

## 4.1. Lower Bound

Here, we outline how constant-width, log-precision transformers can compute L-uniform $TC^0$.[4]

**Intuition: Constant-width PEs.** The key challenge in simulating L-uniform $TC^0$ with constant-width transformers is encoding positional information in constant space; London & Kanade (2025) use logarithmic width to store binary PEs. We use *unit-length* PEs (cf. Lem. B.4) that encode positions in two dimensions while maintaining sufficient separation for attention-based indexing. The encoding maps a position $n$ to $\left( \sqrt{1/n_{tgt}} \quad \sqrt{1 - 1/n_{tgt}} \right)^\top$ where $n_{tgt}$ is the position that the $n^{th}$ symbol has to attend to (e.g., if the $n^{th}$ symbol encodes an input to a gate, $n_{tgt}$ would correspond to the position of that gate). This encoding has approximately unit length (the approximation is due to fixed-point rounding) while ensuring sufficient *gap* between attending to $n_{tgt}$ and $n' \neq n_{tgt}$ to be able to focus on $n_{tgt}$ exclusively. Following London & Kanade (2025), this allows us to encode any L-uniform circuit into the PEs, enabling simulation by a transformer. This contrasts with Merrill & Sabharwal's (2025a) construction, which stores a $F0$-uniform $TC^0$ circuit family (with $F0$-uniform $TC^0 \subseteq$ L-uniform $TC^0$) in the parameters of a fully uniform transformer family. Thus:

**Lemma 4.1.** L-*uniform* $TC^0 \subseteq$ L-*uniform* $LPT^0_{1,c}$.

**Precision compensates for width.** An immediate corollary of Lem. 4.1 is that logarithmic precision can compensate for logarithmic width—that log-precision constant-width transformers ($LPT^0_{1,c}$) are more powerful than those with constant precision but logarithmic width ($LPT^0_{c,1}$):

**Corollary 4.1.** L-*uniform* $LPT^0_{1,c} \supset$ L-*uniform* $LPT^0_{c,1}$.

To the best of our knowledge, this is the first result formally showing that precision can compensate for width in transformer expressivity—for both AHATs and SMATs. While Merrill & Sabharwal's (2025a) results on fully-uniform AHATs lower bound the expressivity of constant-width transformers, they do not address growing width. Moreover, Li et al.'s (2024b) and London & Kanade's (2025) results only consider growing-width transformers, leaving the expressivity of constant-width ones open.

---

[4]We generalize London & Kanade's (2025) constructions rather than using Lem. 3.1 and Merrill & Sabharwal's (2025a) equivalences as lower bounds since Merrill & Sabharwal's (2025a) results concern $F0$-uniform $TC^0$ rather than its superset L-uniform $TC^0$. By extending London & Kanade's (2025) constructions directly, we arrive at the tighter lower bound of L-uniform $TC^0$.

## 4.2. Upper Bounds

The following lemma formalizes that, even with polynomial width, constant-depth transformers cannot exceed L-uniform $\text{AC}^0$ with constant precision and L-uniform $\text{TC}^0$ with growing precision. Intuitively, a constant-depth transformer only performs a constant number of attention operations. Each attention layer computes a weighted average of input representations, which can, even with polynomial resources, be simulated by circuits of constant depth. Polynomial precision and width do not enable the transformer to perform fundamentally more powerful computations—they only provide more space to store intermediate values. Thus:

**Lemma 4.2.** L-*uniform* $\text{LPT}^0_{c,p} \subseteq$ L-*uniform* $\text{AC}^0$ *and* L-*uniform* $\text{LPT}^0_{p,p} \subseteq$ L-*uniform* $\text{TC}^0$.

The separation intuitively arises from the inability to perform unbounded *counting*, which is needed for computing threshold gates with constant-precision activations. Thus, constant-precision transformers remain limited to $\text{AC}^0$.

**The constant-depth picture.** Combining Thm. 4.1, the constant-width lower bound in Lem. 4.1, and the upper bound in Lem. 4.2 closes both sandwiches and yields the headline characterization of the constant-depth regime:

**Theorem 4.2** (Constant-depth expressivity collapse). *As long as the transformer has sufficient volume (Def. 2.3) and width at most polynomial, for any width $D(N)$:*

$$\text{L-}uniform\ \text{LPT}^0_{c,D} = \text{L-}uniform\ \text{AC}^0, \tag{4a}$$

$$\text{L-}uniform\ \text{LPT}^0_{1,D} = \text{L-}uniform\ \text{TC}^0. \tag{4b}$$

Note that the volume constraint imposes a precision-dependent floor on width: In the constant-precision case it requires $D(N) = \Omega(\log N)$, whereas in the log-precision case any width suffices since the precision already supplies $\Omega(\log N)$ volume. Thus, for any volume $V(N) = \Omega(\log N)$, the expressivity of constant-depth L-uniform polynomially padded transformers depends only on whether precision is constant or grows.[5] This is precisely the AC/TC divide marked by the purple lines in Fig. 1: Once volume is sufficient, the only thing that matters for expressivity is which side of the divide the precision falls on.

## 5. Looped Padded Transformers Are Highly Uniform Growing-depth Circuits

We now extend the results to the equivalence between $\Theta(\log^d N)$-*looped* transformers and circuit complexity classes $\text{AC}^d$ and $\text{TC}^d$ for $d \geqslant 1$. As in the constant-depth

regime, we find a *precision-dependent separation*: Constant precision leads to FO-uniform $\text{AC}^d$, and growing precision achieves FO-uniform $\text{TC}^d$. We first discuss a general upper bound for looping circuit layers (§5.1; Lem. 5.1). Perhaps surprisingly, applying this lemma to our constant-depth transformer expressivity results shows that even looped L-uniform transformers are *at most* as powerful as *fully uniform* ones. This then conveniently allows us to apply the *lower bounds* on logarithmic-precision fully-uniform transformers from Merrill & Sabharwal (2025a) to the L-uniform growing-precision case, establishing the equivalence of $\log^d N$-looped padded growing-precision transformers to FO-uniform $\text{TC}^d$. Existing lower bounds, however, do not cover *constant-precision* transformers, which we consider separately. We extend Merrill & Sabharwal's (2025a) lower bounds to the constant-precision case, showing that L-uniform polynomially padded constant-precision transformers are equivalent to FO-uniform $\text{AC}^d$ (§5.2).[6]

### 5.1. A General Upper Bound

We begin with a general lemma showing that looping can be captured by very uniform growing-depth circuit classes, which gives an upper bound on the expressivity of looped transformers. Intuitively, an $r(N)$-looped transformer repeatedly applies the same transformation $f$ to its input, computing $f^{r(N)}$. If $f$ can be computed by a depth-$d(N)$ circuit, then $f^{r(N)} \stackrel{\text{def}}{=} \underbrace{f \circ \cdots \circ f}_{r(N)}$ can be computed by a depth-$r(N) \cdot d(N)$ circuit. For transformers with $r(N) = \Theta(\log^d N)$ loops over $d(N) = \Theta(1)$ layers, this yields depth $\Theta(\log^d N)$.

**Lemma 5.1** (Lem. 9). *Let $m(N)$, $d(N)$, and $r(N)$ be functions at most polynomial in $N$ and logspace-computable given $1^N$ such that $r(N) \cdot d(N) \geqslant 1$. Let $f \colon \{0,1\}^{m(N)} \to \{0,1\}^{m(N)}$ be computed by an L-uniform polynomial-size circuit family of depth $\mathcal{O}(d(N))$ with AND, OR, NOT gates (resp. additionally with THR gates). Then $f^{r(N)}$ is computed by an FO-uniform polynomial-size circuit family of depth $\mathcal{O}(r(N)\,d(N))$ with the same gate set.*

This leads to the following upper bounds:

**Lemma 5.2** (Lem. 6). *For $d \geqslant 1$:*

$$\text{L-}uniform\ \text{LPT}^d_{c,p} \subseteq \text{FO-}uniform\ \text{AC}^d \tag{5a}$$

$$\text{L-}uniform\ \text{LPT}^d_{1,p} \subseteq \text{FO-}uniform\ \text{TC}^d \tag{5b}$$

**Uniformity collapse.** Lem. 5.2 shows that L-uniform looped transformers are upper-bounded by FO-uniform circuit classes. Intuitively, this occurs because the looped

---

[5]Contrast this to polynomially padded *fully* uniform AHATs, equivalent to FO-uniform $\text{TC}^0$ (Merrill & Sabharwal, 2025a).

[6]To facilitate a comparison to Merrill & Sabharwal's (2025a) results, the brackets in our definitions and results in this section list the analog in Merrill & Sabharwal (2025a).

architecture has a fixed set of parameters that are reused at each iteration of the loop and thus result in simpler circuits. More formally, this is a consequence of "*uniformity collapse*" resulting from the fact that $L \subseteq$ FO-uniform $AC^d$ for $d \geqslant 1$, meaning that FO-uniform $AC^d$ circuits can themselves implement any L construction. This contrasts with the constant-depth regime, where L-uniform transformer families achieve L-uniform circuit classes.

## 5.2. Lower Bounds via Circuit Evaluation

**Equivalence for growing-precision transformers.** §5.1 provides an upper bound for L-uniform looped transformers. Merrill & Sabharwal (2025a) show the *matching lower bound* for (fully uniform) growing-precision AHATs. This lower bound naturally applies to L-uniform AHATs, and, using Lem. 3.1, to L-uniform SMATs too. This immediately gives us a complete characterization of growing-precision L-uniform padded looped transformers for $d \geqslant 1$:

$$\text{L-uniform } LPT^d_{1,c} = \text{FO-uniform } TC^d. \tag{6}$$

The rest of the section establishes the matching lower bound for *constant-precision* transformers, showing, for $d \geqslant 1$:

$$\text{L-uniform } LPT^d_{c,1} = \text{FO-uniform } AC^d. \tag{7}$$

We establish the lower bound using a reduction-based technique. The strategy has three steps, each of which adapts a step in Merrill & Sabharwal (2025a) to constant precision.

**Step 1: The reduction framework.** We show that if L-uniform looped constant-precision transformers can recognize a $\mathcal{C}$-complete language $\mathcal{L}$ under $\mathcal{R}$ reductions and can compute all $\mathcal{R}$ reductions, then they can recognize all of $\mathcal{C}$. To do so, we recall Merrill & Sabharwal's (2025a) framework for reductions with transformers.

**Definition 5.1** (Def. 9). *Let $\mathcal{R}$ be a class of languages. $t: \Sigma^* \to \Sigma^*$ is an $\mathcal{R}$ **reduction** if $|t(w)|$ is polynomial in $|w|$ and $\mathcal{L}_t \stackrel{\text{def}}{=} \{(w, B(n), w) \mid t(w)_n = w\}$ is in $\mathcal{R}$.*

**Definition 5.2** (Def. 10). *A transformer family **computes an $\mathcal{R}$ reduction** $t$ if it recognizes the language $\mathcal{L}_t$.*

The following lemma shows that transformers can use reductions to recognize any language in a complexity class if they can recognize a language complete for that class and can compute any relevant reduction.

**Lemma 5.3** (Lem. 3). *Let $\mathcal{C}, \mathcal{R}$ be language classes and let $\mathcal{L}$ be $\mathcal{C}$-complete under $\mathcal{R}$ reductions. If $\mathcal{L} \in$ L-uniform $LPT^d_{c,1}$ and L-uniform $LPT^d_{c,1}$ can compute every $\mathcal{R}$ reduction, then $\mathcal{C} \subseteq$ L-uniform $LPT^d_{c,1}$. The analogous claim holds for L-uniform $LPT^d_{1,c}$.*

Lem. 5.3 allows us to show transformer expressivity lower bounds—for example, that transformers can recognize a

class $\mathcal{C}$—by showing that *(1)* transformers can compute all reductions of a particular complexity class $\mathcal{R}$ (we will consider $\mathcal{R} = L$ reductions in Step 2) and *(2)* transformers can recognize a language $\mathcal{L}$ complete for $\mathcal{C}$ (we will consider the language $\mathcal{L}$ of (wide) $AC^d$ circuit evaluation, which is complete for $\mathcal{C} = AC^d$ under L reductions in Step 3).

**Step 2: Computing L reductions.** We show that *graph connectivity*, which is complete for NL—and thus L—under FO reductions, can be solved by L-uniform $LPT^1_{c,1}$—L-uniform constant-precision log-width transformers with $\Theta(\log N)$ looping. Looped transformers can solve graph connectivity by iteratively computing *reachability*, where each iteration propagates reachability information along the graph's edges. After logarithmically many iterations in the number of vertices, all reachable vertices are discovered; Thm. C.1 formalizes this. Thus, Lem. 5.3 combined with Thm. 4.1, which gives L-uniform $LPT^0_{c,1} =$ L-uniform $AC^0 \supseteq$ FO-uniform $AC^0 =$ FO, yields:

**Lemma 5.4** (Lem. 4). $NL \subseteq$ L-*uniform* $LPT^1_{c,1}$.

Thus, L-uniform $LPT^1_{c,1}$ can compute all $L \subseteq NL$ reductions.

**Step 3: Transformers can evaluate circuits.** Finally, we show L-uniform looped constant-precision transformers can solve wide $AC^d$ circuit evaluation (cf. §C.3.1), which is complete for FO-uniform $AC^d$ under L reductions. This involves the transformer receiving the encoding of a circuit (Def. C.1) and a string to evaluate it on as input, and computing the circuit's output on that string. Intuitively, a transformer can evaluate a circuit by encoding it in its residual stream and evaluating gates level-by-level. In each loop, it identifies gates whose inputs have been computed, evaluates them, and propagates results forward. After $\Theta(L + \log N)$ iterations (where $L$ is the circuit depth and $\log N$ loops are required for circuit pre-processing), the output gate's value is computed. This is formalized in Lem. C.3 and implies:

**Corollary 5.1** (Cor. 7.1). *For $d \geqslant 1$, the wide-$AC^d$ circuit evaluation problem is in L-uniform $LPT^d_{c,1}$.*

From this, we deduce the inclusion of FO-uniform $AC^d$ in L-uniform looped constant-precision transformers. With Lem. 5.2, this characterizes padded looped transformers:

**Theorem 5.1** (Thm. 2). *For any $d \geqslant 1$:*

$$\text{L-}uniform\ LPT^d_{c,1} = \text{FO-}uniform\ AC^d \tag{8a}$$

$$\text{L-}uniform\ LPT^d_{1,c} = \text{FO-}uniform\ TC^d. \tag{8b}$$

## 6. Discussion

**Transformer volume and the interplay of precision and width.** Fig. 1 reveals the importance of *sufficient volume* for padded transformer expressivity: As long as the volume grows logarithmically with string length ($V(N) =$

$\Omega(\log N)$), expressivity only depends on whether or not precision grows—the result of the AC/TC expressivity divide. Constant-precision transformers are constrained to L-uniform $\text{AC}^d$ while growing-precision transformers span L-uniform $\text{TC}^d$ at every depth level $d$. This cannot be compensated for with a polynomial increase in model *width*. We note that the constructions connecting transformers to $\text{TC}^0$ only require growing precision for representing the model *activations*. This aligns with the common techniques of quantizing model weights to lower precision while keeping transformer parameters high-precision (Groeneveld et al., 2024; OpenAI, 2025). Such quantization is sufficient to achieve the entire (L-uniform) $\text{TC}^0$ expressivity.

Padded transformers with *insufficient* volume ($V(N) = o(\log N)$) remain difficult to describe. Existing work describes *unpadded* variants of fully-uniform constant-precision AHATs and SMATs as equivalent to $\text{PFO}^2[<]$—two-variable first-order logic extended with the linear order $<$, a limited fragment of first-order logic (Li & Cotterell, 2025)—but it remains unknown whether padding or looser uniformity increase their expressivity. Crucially, results with $V(N) = o(\log N)$ rely on *causal masking* to provide positional information in the absence of PEs; since constant volume prevents using injective PEs, unmasked attention is particularly limited in this case. In contrast, our constructions are agnostic to the use of masking as, with sufficient volume, masked transformers can be simulated by unmasked ones and vice versa (Merrill & Sabharwal, 2025a, Lem. 1 and Svete & Sabharwal, 2026, Lem. D.14).

**The role of padding.** We conjecture that any "reasonable" parameterization of *unpadded* transformers will be unable to simulate full $\text{AC}^0$ or $\text{TC}^0$ classes.[7] By *reasonable*, we mean transformers with volume $V(N) = \Theta(\log N)$, i.e., models with sufficient volume to capture *entire* $\text{AC}^0$ and $\text{TC}^0$ classes when padded, but whose size scales slowly with $N$.

**Conjecture 6.1.** *Let* $\{\mathcal{T}_N\}_{N\in\mathbb{N}}$ *be an X-uniform unpadded transformer family with* $V(N) = \Theta(\log N)$. *Then,*

*(1) if* $D = \Theta(\log N)$ *and* $b = \Theta(1)$, *there exist X-uniform* $\text{AC}^0$ *circuits that cannot be simulated by the family, and*

*(2) if* $D = \Theta(1)$ *and* $b = \Theta(\log N)$, *there exist X-uniform* $\text{TC}^0$ *circuits that cannot be simulated by the family*

*unless the* X-uniform $\text{AC}^0/\text{TC}^0$ *size hierarchies collapse.*

The intuition behind this conjecture stems from *size hierarchy* results in circuit complexity, which state that entire $\text{AC}^0$ and $\text{TC}^0$ classes cannot be captured by circuits whose size (the number of gates or wires) is bounded by some fixed polynomial. In the non-uniform $\text{AC}^0$ setting, the polynomial-size hierarchy is strict: For any fixed $K \in \mathbb{N}$, there exist

functions in (non-uniform) $\text{AC}^0$ that cannot be computed by circuits of size $\mathcal{O}(N^K)$ (Arora & Barak, 2009). In the $\text{TC}^0$ and uniform $\text{AC}^0$ settings, however, no analogous size hierarchies are known (London & Kanade, 2025). Because any unpadded transformer processing inputs of length $N$ can be simulated by a circuit of size $\mathcal{O}(N^K)$ for some fixed polynomial determined by the transformer's architecture, we can conclude that unpadded non-uniform constant-precision transformers will miss some parts of $\text{AC}^0$ (as already shown by London & Kanade (2025)). Conj. 6.1 extends this separation to uniform $\text{AC}^0$ and $\text{TC}^0$ circuits.

Existing constructions also do not quantify the *minimal* padding required to solve specific problems beyond using $N^K$ padding symbols to compute the values of $N^K$ circuit gates. This further suggests connecting transformers with limited padding to restricted circuit classes such as physically-realizable circuits (Prada & Mali, 2025a;b), whose size cannot scale as an arbitrary polynomial.

It remains open to what degree polynomial model *width* and numeric *precision* can compensate for or even eliminate the need for padding. We argue, however, that those regimes might not be as interesting, since padded transformers more naturally describe *uniform* and *distributed* models of computation. Growing width forces some degree of non-uniformity in the transformer family, while padding constant-width transformers allows for strict parameter uniformity. Simultaneously, padding ensures *distributed* processing of information rather than gathering all information in a few positions of a polynomially-wide transformer.

**The role of uniformity and positional encodings.** Our focus on L-uniform transformer families allows a direct connection to existing results on L-uniform families (London & Kanade, 2025).[8] The power of logspace Turing machines is particularly useful for computing PEs. For example, our constructions that generalize the constant-precision results to looping rely on functions such as division and modulo, which the $\text{AC}^0$-upper-bounded constant-precision transformers cannot compute. In this case, PEs, despite only depending on the position and not the input string, provide the model with information that it cannot itself compute. This suggests that a constant-precision transformer family must have L- (or less) uniform PEs for looping to lead to expected scaling behavior analogous to circuits (i.e., to increase expressivity). In other words, more uniform (and thus simpler) PEs might not contain enough information for constant-precision transformers to benefit from looping.

The definition of transformer family uniformity also allows

---

[7] Note that all equivalences in Fig. 1 naturally apply as *upper bounds* to unpadded transformers.

[8] Note that London & Kanade's (2025) equivalences generalize to any uniformity class as their construction directly uses the computational resources of one computational model (e.g., a circuit) to construct the other one (the equivalent transformer).

us to study the complexity of constructing the transformer parameters and the PEs separately. This is useful, as the interest in length generalization and algorithm learning motivates the study of *fully* uniform families (which reuse identical parameters for all input lengths). While some work is beginning to look at the expressivity enabled by different PEs (Yang et al., 2024; Li et al., 2024a; Li & Cotterell, 2025; 2026), it is an open question how changing the complexity of PEs changes transformer expressivity. Uniform transformer families allow one to study such questions precisely. For example, describing the expressivity of transformers with no or FO-uniform PEs would quantify the importance and the expressivity gain afforded by L-uniform PEs.

**Looping vs. depth.** The distinction between *looping* and *depth* matters because the two scale expressivity through different mechanisms: A *L-deep* transformer has $L$ *independent* sets of parameters, so increasing depth grows both the computation graph *and* the parameter count, while a *L-looped* transformer reuses a single fixed set of layers $L$ times, growing the computation graph without growing the parameter count and yielding a more uniform computational model. Our results nevertheless carry over to deep transformers. Since any $L$-looped transformer is a special case of an $L$-deep one (with all layers tied), our lower bounds—constructions of FO-uniform $\mathsf{AC}^d$ and FO-uniform $\mathsf{TC}^d$ circuits—transfer to $\Theta(\log^d N)$-deep L-uniform transformers. The matching upper bounds transfer as well: A $\Theta(\log^d N)$-deep L-uniform family has a number of *independent* layers that grows with $N$, but for the family to be L-uniform the logspace construction machine must itself emit the parameters of all $\Theta(\log^d N)$ layers from input $1^N$. Then, due to uniformity collapse (cf. §5.1), the same FO-uniform $\mathsf{AC}^d$ and FO-uniform $\mathsf{TC}^d$ upper bounds apply here. Looping is thus the more parameter-efficient—and, arguably, the more natural—way of obtaining the scaling behavior we describe, but the same expressivity is recovered if the additional layers carry independent parameters.

**Resource allocation.** Our results suggest expressivity-driven *resource allocation* in real-world transformers. Model depth and precision increase expressivity, but both incur a degree of non-parallelizable *sequentiality*. However, expressivity *scales better with depth than with precision*: Given a budget of, e.g., $\log^d N$ sequential steps, our results suggest devoting only $\log N$ of those to numeric precision and the remaining $\log^{d-1} N$ ones to looping, since additional precision would not increase expressivity, whereas expressivity scales gracefully with looping. We note, however, that we only consider expressivity; while different uniformity regimes have implications for generalization, we do not make any claims about the learnability of the considered classes of languages. A growing body of literature considers this question both theoretically (Hahn & Rofin,

2024; Yang et al., 2025; Merrill et al., 2026; Kövér et al., 2026, *inter alia*) and practically (Weiss et al., 2018; Bhattamishra et al., 2020; van der Poel et al., 2024; Delétang et al., 2023; Someya et al., 2024; Borenstein et al., 2024; Svete et al., 2024; Butoi et al., 2025, *inter alia*); the connection of model uniformity to learnability is an exciting avenue for future work.

**The role of transformer arithmetic.** Throughout the paper, we use **rounding** of fixed-point arithmetic results as a "hammer" that allows transformers to perform useful gadgets (e.g., focus on individual positions). Specifically, under fixed-point arithmetic, every arithmetic operation is followed by a projection onto the closest representable value in $\mathbb{F}_b$ (cf. §A.3), which lets us implement non-linear behavior— e.g., thresholding and exact comparisons—inside the otherwise affine layers of a transformer. This is true for *both* constant and growing precision: In either regime, rounding turns small numerical gaps into sharp combinatorial decisions, and our constructions rely on choosing weights large enough that the gap between attended and non-attended positions is wider than the rounding error. The fact that rounding applies across different transformer idealizations is part of why our results compose cleanly across constant and growing precision. Note that our results (particularly upper bounds) may not transfer to *floating point* transformers (Li et al., 2024b; Park et al., 2026).

# 7. Conclusion

We show that polynomially padded transformers align well to natural circuit complexity classes and are surprisingly robust to variations in attention type, model width, and uniformity—as long as the model has enough memory per symbol to index individual positions. This contrasts them with unpadded transformers, whose expressivity is difficult to link to well-known circuit classes and is brittle in parametrization. Concretely, L-uniform padded constant-precision transformers span L-uniform $\mathsf{AC}^0$ while growing-precision ones reach L-uniform $\mathsf{TC}^0$. Moreover, they scale analogously to circuits when looped, showing that looping is a viable way of increasing inference-time compute by increasing model expressivity. The most pressing question they leave open is a tight characterization of *insufficient-volume* ($V(N) = o(\log N)$) transformers, as standard $\mathsf{AC}^0/\mathsf{TC}^0$ classes appear too coarse and likely require sub-$\mathsf{AC}^0$ classes such as $\mathsf{PFO}^2[<]$. Altogether, these results establish a more unified view of transformer expressivity and solidify its connection to natural circuit classes.

# Impact Statement

This paper presents work whose goal is to advance the field of machine learning. There are many potential societal

consequences of our work, none of which we feel must be specifically highlighted here.

## Acknowledgments

Anej Svete is supported by the ETH AI Center doctoral fellowship. Many ideas and the motivation for this work were first developed during the 2025 Dagstuhl Seminar 25282 "Theory of Neural Language Models" (Barceló et al., 2026), particularly in the "Uniformity" working group attended by Satwik Bhattamishra, Michaël Cadilhac, David Chiang, Will Merrill, Ashish Sabharwal, Clayton Sanford, Howard Straubing, Laura Strieker, and Anej Svete. We thank the attendees for insightful discussions. We also thank Reda Boumasmoud for helpful discussions about fixed-point arithmetic.

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

# Contents of the Appendix

# A. Preliminaries

## A.1. Notation

Let $\Sigma$ be an alphabet, a finite, non-empty set of **symbols**. A **language** $\mathcal{L}$ is a subset of $\Sigma^* \stackrel{\text{def}}{=} \bigcup_{N \in \mathbb{N}} \Sigma^N$, the set of all strings. We denote the concatenation of two strings $w_1, w_2 \in \Sigma^*$ as $w_1 \circ w_2$ or simply $w_1 w_2$. We use $\texttt{poly}(N) \stackrel{\text{def}}{=} \{f \colon \mathbb{N} \to \mathbb{N} \mid \exists K > 0, f(N) = \mathcal{O}(N^K)\}$ to denote the set of functions with at most polynomial growth rate.

## A.2. Circuit Complexity

Boolean circuits are a model of parallel computation that processes binary input strings through a series of logical operations to produce binary outputs.[9] Formally, a **boolean circuit** is a directed acyclic graph where source nodes represent the $N$-bit input, and a single sink node represents the output. Non-source vertices are called **gates** and are labeled with logical operations (e.g., AND, OR, NOT). The **size** of a circuit is the number of gates, and its **depth** is the longest path from any input to the output.

A circuit $C_N$ computes a function $C_N \colon \{0,1\}^N \to \{0,1\}$ for some $N \in \mathbb{N}$; we attach the subscript $N$ to be compatible with the circuit-family notation $\{C_N\}_{N \in \mathbb{N}}$ used below. The value $C_N(w)$ for input string $w \in \{0,1\}^N$ is computed by evaluating the gates in topological order starting from the input bits. We say that the circuit $C_N$ **accepts** a string $w$ if $C_N(w) = 1$.

**Circuit families** process input strings of variable length. A circuit family is a sequence of circuits $\mathcal{C} \stackrel{\text{def}}{=} \{C_N\}_{N \in \mathbb{N}}$ where $C_N$ processes inputs of length $N$. A circuit family is said to recognize a language if, for any given input string, the corresponding circuit outputs 1 if and only if the string is in the language.

A **circuit complexity class** is a set of circuit families that satisfy certain constraints on size, depth, and the types of gates used. This paper focuses on two common classes:

- $\text{AC}^{\text{d}}$: Circuits with NOT, AND, and OR gates that have unbounded fan-in, size polynomial in $N$, and depth $\mathcal{O}(\log^d N)$.
- $\text{TC}^{\text{d}}$: The extension of $\text{AC}^{\text{d}}$ that adds **threshold gates** THR, which output 1 if the sum of their inputs exceeds a given threshold. It is known that $\text{AC}^0 \subsetneq \text{TC}^0$ and $\text{AC}^{\text{d}} \subseteq \text{TC}^{\text{d}}$. For example, PARITY, the language of binary strings with an even number of 1s, is in $\text{TC}^0$ but not in $\text{AC}^0$ (Furst et al., 1984).

Without additional constraints, circuit families can recognize undecidable languages by having arbitrary solutions for each input length. To avoid this and ensure the model of computation is realistic, we can impose a **uniformity** condition. A circuit family is **uniform** if there exists a Turing machine that, on input $1^N$, generates a description of the circuit $C_N$. In particular, a circuit class is **L-uniform** if a Turing machine using $\mathcal{O}(\log N)$ space can construct its description from the input $1^N$. This ensures the circuits for different input lengths are related by a systematic procedure. A finer notion of uniformity, used throughout our main results, is **FO-uniformity**. To state it, fix a standard encoding of the gates of $C_N$ by binary strings of length $\mathcal{O}(\log N)$; the **connection language** of the family $\{C_N\}_{N \in \mathbb{N}}$ then consists of the tuples $(1^N, \text{B}(i), \text{B}(j), t)$ such that gate $i$ in $C_N$ has type $t$ (input, AND, OR, NOT, THR, or output) and is wired to gate $j$. A circuit family is **FO-uniform** if its connection language is definable in first-order logic over strings with the predicates $+$, $\times$, and $\leqslant$ on position indices (Mix Barrington et al., 1990); equivalently, if a DLOGTIME Turing machine decides it. Because FO $\subseteq$ L, every FO-uniform family is L-uniform.

## A.3. Fixed-point Arithmetic

Our computation models perform operations with **fixed-point arithmetic** (Li et al., 2024b; Saunshi et al., 2025; London & Kanade, 2025; Svete & Sabharwal, 2026).

**Definition A.1** (Fixed-point representation)**.** *Let $b \in \mathbb{N}$ be the number of bits devoted to each of the integer and fractional parts. We use $\mathbb{F}_b$ to denote the set*

$$\mathbb{F}_b \stackrel{\text{def}}{=} \{x_\pm \cdot a \cdot 2^{-b} \mid x_\pm \in \{-1, 1\}, a \in \{0, 1, \dots, 2^{2b} - 1\}\} \tag{9}$$

We define $B_{\mathbb{F}} \stackrel{\text{def}}{=} \max \mathbb{F}_b = 2^b - 2^{-b}$. All values exceeding $B_{\mathbb{F}}$ are considered out of range and are rounded to $B_{\mathbb{F}}$. Note, however, that $B_{\mathbb{F}}$ does *not* behave like infinity—it is not an annihilator (in the algebraic sense), i.e., does not absorb all

---

[9]By representing symbols from any alphabet with binary encodings, circuits (or circuit functions) can be used to process strings over any finite alphabet. We focus on binary strings for simplicity.

subsequent operations. For example, for any non-negative $x \in \mathbb{F}_b$, $B_\mathbb{F} - x \neq B_\mathbb{F}$ is a valid number. To handle the results of arithmetic operations that may not be exactly representable in the fixed-point format, we define a standard for rounding.

**Definition A.2** (Rounding). *For any $x \in \mathbb{R}$ and any closed subset $\mathbb{F}$ of $\mathbb{R}$ containing 0, we define* $\mathsf{round}(x, \mathbb{F})$ *as the closest number to $x$ in $\mathbb{F}$. In case of a tie, the value with the larger absolute value is chosen.*

We denote the rounding operation as $[\cdot]_b \overset{\text{def}}{=} \mathsf{round}(\cdot, \mathbb{F}_b)$. This operation is applied to vectors and matrices element-wise. All binary operations are defined by first performing the ideal mathematical operation and then rounding the result to the nearest representable value in $\mathbb{F}_b$. Division by zero is considered an error and results in an incorrect output.

For operations involving more than two numbers, rounding is applied iteratively.

**Definition A.3** (Summation with iterative rounding). *For $b, N \in \mathbb{N}$ and $\boldsymbol{x} \in \mathbb{R}^N$, we define summation with iterative rounding to $b$ fractional bits as the function* $\mathsf{SUM}_b \colon \bigcup_{N \in \mathbb{N}} (\mathbb{F}_b)^N \to \mathbb{F}_b$, *where for any $N \in \mathbb{N}^+$ and $\boldsymbol{x} \in (\mathbb{F}_b)^N$:*

$$\mathsf{SUM}_b(\boldsymbol{x}) \overset{\text{def}}{=} \left[\dots \left[[x_1 + x_2]_b + x_3\right]_b + \dots + x_N\right]_b \tag{10}$$

This iterative rounding process is not associative, and the order of operations can affect the final result. Based on this, we can also define more complex operations such as the **fixed-point inner product** $\langle \boldsymbol{x}, \boldsymbol{y} \rangle_b \overset{\text{def}}{=} \mathsf{SUM}_b(\boldsymbol{x} \odot \boldsymbol{y})$, where $\odot$ denotes the element-wise product of two vectors, and **fixed-point matrix product** for matrices $\boldsymbol{A}$ and $\boldsymbol{B}$, where $(\boldsymbol{A} \times_b \boldsymbol{B})_{i,j} \overset{\text{def}}{=} \langle (\boldsymbol{A}_{i,:})^\top, \boldsymbol{B}_{:,j} \rangle_b$. These operations will be used by fixed-point arithmetic transformers; throughout, the rounding step $[\cdot]_b$ that appears in every binary operation and in the iterative summation above is applied after every (sub-)step of the operation in transformer's computation.

## A.4. Transformers and Transformer Families

### A.4.1. THE TRANSFORMER ARCHITECTURE

For a fixed input length $N \in \mathbb{N}$ and model **width** $D \in \mathbb{N}$, a transformer $\mathcal{T}_N$ consists of:

(1) a **symbol embedding** $\boldsymbol{e} \colon \Sigma \to \mathbb{F}^D$ for $w \in \Sigma$,
(2) a **positional encoding** (PE) $\boldsymbol{p} \colon \mathbb{N} \times \mathbb{N} \to \mathbb{F}^D$,
(3) $L$ **layers** $\boldsymbol{\tau}^{(1)}, \dots, \boldsymbol{\tau}^{(L)}$, each of which consists of two sub-layers: A self-attention layer and a position-wise feed-forward network $\boldsymbol{f}$, and
(4) a classification **output layer** $\boldsymbol{o}$ of the form $\boldsymbol{o} \colon \mathbb{F}^D \to \{0, 1\}$.

A transformer with layers $\boldsymbol{\tau}^{(1)}, \dots, \boldsymbol{\tau}^{(L)}$ computes $\boldsymbol{h}_n^{(l)} \in \mathbb{F}^D$ for $l \in \{1, \dots, L\}$ and each position $n \in [N]$ in the input string $\boldsymbol{w} = w_1 \cdots w_N \in \Sigma^*$ as follows:[10]

$$\boldsymbol{h}_n^{(0)} \overset{\text{def}}{=} \boldsymbol{e}(w_n) + \boldsymbol{p}(n, N) \in \mathbb{F}^D \text{ for } n \in [N] \tag{11a}$$

$$\boldsymbol{H}^{(l)} \overset{\text{def}}{=} \left(\boldsymbol{h}_1^{(l)\top} \quad \cdots \quad \boldsymbol{h}_N^{(l)\top}\right)^\top \in \mathbb{F}^{N \times D} \tag{11b}$$

$$\boldsymbol{Q}^{(l)} \overset{\text{def}}{=} \boldsymbol{H}^{(l)} \boldsymbol{W}_Q^{(l)}, \quad \boldsymbol{K}^{(l)} \overset{\text{def}}{=} \boldsymbol{H}^{(l)} \boldsymbol{W}_K^{(l)}, \quad \boldsymbol{V}^{(l)} \overset{\text{def}}{=} \boldsymbol{H}^{(l)} \boldsymbol{W}_V^{(l)} \quad \in \mathbb{F}^{N \times D} \tag{11c}$$

$$\boldsymbol{G}^{(l)} \overset{\text{def}}{=} \boldsymbol{\nu}(M(\boldsymbol{\Delta}(\boldsymbol{Q}^{(l)} \boldsymbol{K}^{(l)\top})) \boldsymbol{V}^{(l)}) + \boldsymbol{H}^{(l)} \in \mathbb{F}^{N \times D} \tag{11d}$$

$$\boldsymbol{H}^{(l+1)} \overset{\text{def}}{=} \boldsymbol{G}^{(l)} + \boldsymbol{\nu}(\boldsymbol{f}(\boldsymbol{G}^{(l)})) \in \mathbb{F}^{N \times D} \tag{11e}$$

Here, $\boldsymbol{\Delta} \colon \mathbb{F}^* \to \mathbb{F}^*$ is a length-preserving function that computes attention weights and is applied row-wise, and $\boldsymbol{\nu} \colon \mathbb{F}^D \to \mathbb{F}^D$ is the layer normalization function (Ba et al., 2016) applied position-wise.[11] Here, $M \colon \mathbb{F}^{N \times N} \to (\mathbb{F} \cup \{-\infty\})^{N \times N}$ is the **masking function**.[12] We say that the $l^{\text{th}}$ layer $\boldsymbol{\tau}^{(l)}$ computes the function $\boldsymbol{\tau}^{(l)} \colon \mathbb{F}^{N \times D} \to \mathbb{F}^{N \times D}$, defined as $\boldsymbol{\tau}^{(l)} \colon \boldsymbol{H}^{(l-1)} \mapsto \boldsymbol{H}^{(l)}$ for $l \in \{1, \dots, L\}$. We also denote with $\mathcal{T}$ the function $\mathcal{T} \colon \Sigma^* \to \mathbb{F}^{N \times D}$, defined as $\mathcal{T} \colon \boldsymbol{w} \mapsto \boldsymbol{H}^{(L)}$.

---

[10]We focus on transformers with a single head per layer. As multi-head attention can be simulated by single-head attention with a constant factor increase in depth (Yang et al., 2026b), our results extend to multi-head transformers as well.

[11]Following previous work, we assume that layer normalization can be applied to *parts* of the vector individually to enable gadgets such as the **layer hash norm** (Merrill & Sabharwal, 2024; 2025a).

[12]Following Li et al. (2024b) and Saunshi et al. (2025), we define masking with a function rather than an additive matrix since subtracting $B_\mathbb{F}$ from an arbitrary number in $\mathbb{F}$ does not necessarily result in $-B_\mathbb{F}$.

Both attention variants studied in theoretical literature are special cases of **temperature-scaled softmax attention**:

$$\boldsymbol{\Delta}(\boldsymbol{x})_n \overset{\text{def}}{=} \text{softmax}_\tau(\boldsymbol{x})_n = \frac{\exp(x_n/\tau)}{\sum_{n'=1}^N \exp(x_{n'}/\tau)} \text{ for } \boldsymbol{x} \in \mathbb{R}^N, n \in [N], \text{ and } \tau > 0. \tag{12}$$

The **temperature** $\tau = \tau(N) > 0$ is a per-model parameter that may depend on the input length and is included alongside the parameter matrices in the description of $\mathcal{T}_N$ produced by the construction machine $\mathcal{M}_1$ of Def. A.5; in particular, $\tau(N)$ must be computable within the same complexity class as the rest of the model. We view this as a natural strengthening of the standard definition: Practical implementations apply softmax to scaled scores ($\boldsymbol{q}^\top \boldsymbol{k}/\sqrt{D}$, with the scaling absorbable into a temperature), and a length-dependent temperature is the standard mechanism by which SMATs approximate AHATs (Yang et al., 2026a).

(1) **Softmax attention** (with a general function $\tau$): We call such transformers **softmax-attention transformers** (SMATs).
(2) **Average hard attention** ($\tau \to 0$): The limit yields

$$\lim_{\tau \to 0} \text{softmax}_\tau(\boldsymbol{x})_n = \begin{cases} \frac{1}{|\arg\max(\boldsymbol{x})|} & \text{if } x_n = \max(\boldsymbol{x}), \\ 0 & \text{otherwise}. \end{cases} \tag{13}$$

We call such transformers **average hard-attention transformers** (AHATs).

**Fixed-point arithmetic transformers.** All transformer operations are performed using fixed-point arithmetic from §A.3 for some precision $b$, which can depend on the input length $N$. We focus on three regimes:

- **Constant precision**: $b = \Theta(1)$.
- **Logarithmic precision**: $b = \Theta(\log N)$.
- **Polynomial precision**: $b = \texttt{poly}(N)$.

We also allow our transformers to use *higher* precision when computing individual components, such as the attention sub-layer or the position-wise MLP: A component may perform its internal arithmetic at a precision $b' \geqslant b$ as long as $b' = \Theta(b)$ and its output is then *clamped* back to the residual-stream precision $\mathbb{F}_b$ when written between components. This allows, for instance, the attention weights—which can be sensitive to high or low activations—to be computed more precisely, and the construction of equivalent transformers to manipulate intermediate quantities (such as rescaled query matrices) that may not be exactly representable in $\mathbb{F}_b$. At the granularity of the precision regimes we consider (constant, logarithmic, or polynomial precision), this mixed-precision allowance does *not* change the expressivity of the transformer—since the residual-stream precision $b$ remains the regime's defining quantity, and $b' = \Theta(b)$ stays inside the same regime—but it considerably simplifies some constructions by removing the need to fit auxiliary computations exactly into $\mathbb{F}_b$. This aligns with the modern practice of training large models with quantized weights while allowing certain activations (such as the attention weights) to use higher precision (Groeneveld et al., 2024; Merrill & Sabharwal, 2025b).

$b$-**precision.** Many existing expressivity results rely on the exact nature of the arithmetic used. Merrill & Sabharwal (2025b) (and subsequent work) abstract the datatype, requiring only that the operations are $b$-precise in the following sense.

**Definition A.4.** *Let $f : \mathbb{R}^N \to \mathbb{R}$ be a function with the $\mathbb{F}_b$ realization $\widetilde{f} : \mathbb{F}_b^N \to \mathbb{F}_b$. We say that $\widetilde{f}$ is $b$-precise if, for any $x_1, \dots, x_N \in \mathbb{F}_b$,*

$$\text{round}(f(x_1, \dots, x_N)) = \widetilde{f}(x_1, \dots, x_N). \tag{14}$$

By definition of rounding to the nearest representable number, the components of a transformer are $b$-precise. Moreover, for all growing precision ($b(N) = \Omega(\log N)$), attention is also $b$-precise (Merrill & Sabharwal, 2025b).[13]

Fixed-point arithmetic thus both limits what can be computed and enables the construction of various gadgets that leverage the arithmetic to implement logical operations and attention patterns. We collect some useful gadgets in §B. The proofs of our main results then use these constructions to implement the necessary computations for simulating circuit classes with transformers and vice versa.

---

[13]Since we derive all results for constant-precision transformers without relying on $b$-precision, we do not require it from constant-precision transformers. $b$-precision of growing-precision transformers, however, allows us to apply all results from Merrill & Sabharwal (2025b) and Merrill & Sabharwal (2025a) to that case.

A.4.2. TRANSFORMER FAMILIES AND UNIFORMITY

Analogously to circuit families, each string length $N$ is processed by a separate transformer. To process all of $\Sigma^*$, we therefore define a **transformer family** $\{\mathcal{T}_N\}_{N \in \mathbb{N}}$ as a sequence of transformers where each $\mathcal{T}_N$ processes strings of length $N$. Further, we impose a uniformity condition on the family, which will relate the transformers for different input lengths.

**Definition A.5** (Uniform transformer families; variant of London & Kanade, 2025, Def. 3.6). *Let* X *be a computational complexity class. A transformer family* $\{\mathcal{T}_N\}$ *is* X-***uniform*** *if there exist Turing machines* $\mathcal{M}_1$ *and* $\mathcal{M}_2$ *whose resource usage is constrained by the complexity class* X *such that:*

*(1)* $\mathcal{M}_1$ *takes input* $1^N$ *and outputs a description of* $\mathcal{T}_N$, *and*
*(2)* $\mathcal{M}_2$ *takes input* $(1^N, \mathtt{B}(n))$ *and outputs* $\boldsymbol{p}(n, N)$.

Def. A.5 allows for size-dependent transformers while keeping them related (as the same Turing machines must construct them for all $N$). It also facilitates natural connections to uniform circuit classes (cf. §A.2) (London & Kanade, 2025). All our results concern L-uniform transformer families, in which case, the Turing machines in Def. A.5 operate in logspace.

**Information-rich positional encodings.** Def. A.5 defines two components of a transformer based on the notion of uniform computability: The transformer model itself and the PE. Although superficially similar, these two components appear in distinctly different roles in our constructions. The transformer model intuitively defines the "algorithm" that processes the input string and is limited in terms of the operations to those implementable by the attention mechanism. For example, although it is *constructed* by a logspace Turing machine, it cannot *compute* arbitrary logspace functions. The PEs, in contrast, provide a way to inject additional information into the model that it cannot compute itself *on the fly*. They can, for example, provide direct access to the binary representation of the position index $n$, and pre-compute arithmetic operations such as division and modulo. This way, PEs provide the transformer with information that it cannot necessarily compute itself, but that is still computable in logspace.

**Language recognition.** We treat a transformer $\mathcal{T}$ as a **language encoder**—a length-preserving function $\Sigma^* \to (\mathbb{F}^D)^*$ (Cotterell et al., 2024; Chan et al., 2024)—and regard the output $\boldsymbol{H}^{(L)}$ on a string $\boldsymbol{w}$ as a $|\boldsymbol{w}| \times D$ matrix, where each row corresponds to the contextual representation of the symbol at the corresponding position. To convert this into a language recognizer, we use the output layer $\boldsymbol{o}$ that maps the contextual representation of the *final symbol* to the membership (classification) decision 0 or 1.

## A.5. Looped Padded Transformers

Looped (or universal) transformers use a fixed block of transformer layers that is applied repeatedly to the input string (Dehghani et al., 2019; Giannou et al., 2023; Hao et al., 2024; Goyal et al., 2024; Chen et al., 2025; Zeng et al., 2026; Geiping et al., 2025). This increases the depth of the model, enabling more complex reasoning by applying layers multiple times, and does not increase the model size, as the same block is reused for each iteration, thus reducing the memory footprint and computational cost (Bae et al., 2026). We define looped transformers as follows.

**Definition A.6** (Looped transformer). *Let* $L, T \in \mathbb{N}$ *and let* $1 \leqslant l_1 \leqslant l_2 \leqslant L$. *Given a depth-$L$ transformer, a* ***looped transformer*** *computes symbol contextual representations* $\boldsymbol{H}$ *by*

1. *Computing the initial hidden states* $\boldsymbol{H}^{(0)}$ *for the input string* $\boldsymbol{w} = w_1 \cdots w_N$ *and computing* $\boldsymbol{H}^{(l_1)}$ *using the first* $l_1$ *layers of the transformer.*
2. *Applying the transformer layers* $l_1 + 1, \ldots, l_2$ *$T$ times to the hidden states* $\boldsymbol{H}^{(l_1)}$ *to obtain* $\boldsymbol{H}^{(l_1 + T(l_2 - l_1))}$.
3. *Applying the transformer layers* $l_2 + 1, \ldots, L$ *to the hidden states* $\boldsymbol{H}^{(l_1 + T(l_2 - l_1))}$ *to obtain the final representations* $\boldsymbol{H}$ *that are passed to the output layer.*

The dynamic computational depth of looped transformers lets them perform more complex reasoning tasks by iteratively refining their hidden states across multiple timesteps. Importantly, these reasoning steps combine sequential and parallel processing of the input symbols, yielding both parallel efficiency and reasoning depth. For brevity, Def. A.6 states a single looped block, but the definition extends in the natural way to a finite number of sequentially composed looped blocks $[l_1^{(k)}, l_2^{(k)}]$ with their own iteration counts $T^{(k)}$; this form resembles learned by program-of-layers architectures (Li et al., 2026) and the form we use in our constructions when convenient.

Padded transformers additionally pad the input string with padding (pause) symbols (Pfau et al., 2024; Goyal et al., 2024;

London & Kanade, 2025). The number of padding symbols can depend on the input length.

**Definition A.7** (Padded Transformer)**.** *Given a (looped) transformer $\mathcal{T}$ and a **padding length function** $P\colon \mathbb{N} \to \mathbb{N}$, a **padded transformer** is the pair $(\mathcal{T}, P)$ that computes the contextual representations $\boldsymbol{H}$ of a string $\boldsymbol{w} \in \Sigma^*$ by running $\mathcal{T}$ on the padded input $\boldsymbol{w} \circ \underbrace{\square \cdots \square}_{P(|\boldsymbol{w}|)}$, where $\square \notin \Sigma$ is a designated padding symbol.*

Instead of being restricted to the contextual representations of the $N$ input symbols, a padded transformer can determine string membership or symbol probabilities based on the contextual representations of the $P(N)$ additional padded symbols as well. This additional space can be used to perform more operations and is analogous to increasing the circuit width in circuit complexity.

# B. Theoretical Gadgets

This section contains various (known) theoretical gadgets that we use in the proofs of the main results. In the following, $N \in \mathbb{N}$ always refers to the length of the original input string. We define the **interleaving** of the vectors $\boldsymbol{x}, \boldsymbol{y} \in \mathbb{R}^D$ as $\boldsymbol{x} \frown \boldsymbol{y} \in \mathbb{R}^{2D}$ where

$$\boldsymbol{x} \frown \boldsymbol{y}_d \stackrel{\text{def}}{=} \begin{cases} x_{(d+1)/2} & \textbf{if } d \text{ is odd,} \\ y_{d/2} & \textbf{otherwise.} \end{cases} \tag{15}$$

The following lemma, due to Merrill & Sabharwal (2023, Lem. 1) (where it was adapted from Hao et al. (2022)) and used as Lem. C.2 by London & Kanade (2025), lets us turn any logspace-computable function into an L-uniform $\mathsf{AC}^0$ circuit.

**Lemma B.1** (Merrill & Sabharwal, 2023, Lem. 1)**.** *Let $f\colon \{0,1\}^* \to \{0,1\}^m$ be a function computable in linear space. Then, for any constant $c \in \mathbb{R}_{>0}$, there is a Turing machine that, on input $1^N$, uses at most $c \log N + \log m$ space to output the description of a depth-3 $\mathsf{AC}^0$ circuit of size at most $N^c + c \log N + m$ computing $f$ on inputs of size at most $c \log N$.*

## B.1. Positional Encodings

The following lemmata follow from the definition of fixed-point arithmetic and the rounding and thresholding applied therein.

**Lemma B.2.** *Let $x \in \mathbb{F}_b$ for some $b \in \mathbb{N}$ such that $x > \log 2(b+1)$. Then, it holds that*

$$\exp(x) = B_{\mathbb{F}}, \tag{16a}$$

$$\exp(-x) = 0. \tag{16b}$$

**Lemma B.3** (Generalization of Li et al., 2024b, Lem. E.3)**.** *For $N \in \mathbb{N}$, $n \in [N]$, define the vectors $\boldsymbol{q}_n \in \mathbb{R}^{2\lceil \log N \rceil}$ and $\boldsymbol{k}_n \in \mathbb{R}^{2\lceil \log N \rceil}$ as follows:*

$$\boldsymbol{q}_n \stackrel{\text{def}}{=} B_{\mathbb{F}}/m \cdot \left(\mathsf{B}^{\pm}(n_{tgt}) \frown \boldsymbol{1}_{\lceil \log N \rceil}\right) \tag{17a}$$

$$\boldsymbol{k}_{n'} \stackrel{\text{def}}{=} \mathsf{B}^{\pm}\left(n'\right) \frown \left(-\boldsymbol{1}_{\lceil \log N \rceil}\right). \tag{17b}$$

*Then, it holds that*

$$\langle \boldsymbol{q}_n, \boldsymbol{k}_{n'} \rangle_b = \begin{cases} 0 & \textbf{if } n' = n_{tgt} \\ x & \textbf{otherwise.} \end{cases} \tag{18}$$

*where $x \leqslant -2B_{\mathbb{F}}/m$.*

**Lemma B.4** (Constant-sized positional encodings)**.** *Let $N \in \mathbb{N}$, $b \geqslant 5 \log N$, and $\mu\colon [N] \to \mathbb{F}_b^2$ be defined by*

$$\mu\colon n \mapsto \begin{pmatrix} \sqrt{\frac{1}{n}} \\ \sqrt{1 - \frac{1}{n}} \end{pmatrix} \tag{19}$$

*where all operations are computed in b-bit fixed-point arithmetic. Then, for all $n, n' \in [N]$ with $n \neq n'$:*

$$\langle \mu(n), \mu(n) \rangle_b - \langle \mu(n), \mu(n') \rangle_b \geqslant \frac{1}{2N^4} - C\sqrt{N} \cdot 2^{-b} \geqslant \frac{1}{4N^4}, \tag{20}$$

*where $C > 0$ is an absolute constant.*

*Proof.* By Merrill & Sabharwal (2024, Lem. 8), in exact arithmetic:

$$\mu(n)^\top \mu(n) - \mu(n)^\top \mu(n') \geqslant \frac{1}{2\max(n, n')^4} \geqslant \frac{1}{2N^4}. \tag{21}$$

We now bound the error introduced by fixed-point computation. Let $\mathsf{round}(\cdot)$ denote rounding to $b$ bits of precision, satisfying $|\mathsf{round}(x) - x| \leqslant 2^{-b-1} \overset{\text{def}}{=} \zeta$. We compute $\langle \mu(n), \mu(n') \rangle_b$ for $n, n' \in [N]$ and track errors through each operation. Let $a = \mathsf{round}(\frac{1}{n})$ and $a' = \mathsf{round}(\frac{1}{n'})$, so $|a - \frac{1}{n}| \leqslant \zeta$ and $|a' - \frac{1}{n'}| \leqslant \zeta$.

**Square root.** For $x \in [\delta, 1]$ with $\delta > 0$, the function $\sqrt{x}$ has derivative $\frac{1}{2\sqrt{x}} \leqslant \frac{1}{2\sqrt{\delta}}$. By the mean value theorem, if $|x' - x| \leqslant \epsilon$, then $|\sqrt{x'} - \sqrt{x}| \leqslant \frac{\epsilon}{2\sqrt{\delta}}$. Since $n \geqslant 1$, we have $\frac{1}{n} \in (0, 1]$ and $1 - \frac{1}{n} \in [0, 1)$. For $n \geqslant 2$, both arguments to $\sqrt{\cdot}$ are bounded away from 0 by at least $1/N$. For $n = 1$, we have $\mu(1) = \begin{pmatrix} 1 & 0 \end{pmatrix}^\top$ exactly. Thus, for $n \geqslant 2$:

- Let $b = \mathsf{round}(\sqrt{a})$. Then $|b - \sqrt{\frac{1}{n}}| \leqslant \frac{1}{2\sqrt{1/N}} \cdot \zeta + \zeta = (\frac{\sqrt{N}}{2} + 1)\zeta$.
- Let $c = \mathsf{round}(1 - a)$. Then $|c - (1 - \frac{1}{n})| \leqslant 2\zeta$.
- Let $d = \mathsf{round}(\sqrt{c})$. Then $|d - \sqrt{1 - \frac{1}{n}}| \leqslant \frac{\sqrt{N}}{2} \cdot 2\zeta + \zeta = (\sqrt{N} + 1)\zeta$.

**Inner product.** The inner product $\langle \mu(n), \mu(n') \rangle_b = b \cdot b' + d \cdot d'$ involves multiplying values in $[0, 1]$. For $x, y, x', y' \in [0, 1]$:

$$|x'y' - xy| = |x'y' - x'y + x'y - xy| \leqslant |x'||y' - y| + |y||x' - x| \leqslant |y' - y| + |x' - x|. \tag{22}$$

Combining all error terms and accounting for final rounding in the multiplications and addition, there exists an absolute constant $C' > 0$ such that

$$\left| \langle \mu(n), \mu(n') \rangle_b - \mu(n)^\top \mu(n') \right| \leqslant C'\sqrt{N} \cdot \zeta \tag{23}$$

for all $n, n' \in [N]$.

**Combining the bounds.** For $n = 1$, the arithmetic is exact and the bounds hold trivially. For $n \geqslant 2$: In the worst case, rounding decreases the self-product and increases the cross-product:

$$\langle \mu(n), \mu(n) \rangle_b \geqslant \mu(n)^\top \mu(n) - C'\sqrt{N} \cdot \zeta = 1 - C'\sqrt{N} \cdot \zeta, \tag{24a}$$

$$\langle \mu(n), \mu(n') \rangle_b \leqslant \mu(n)^\top \mu(n') + C'\sqrt{N} \cdot \zeta. \tag{24b}$$

Therefore,

$$\langle \mu(n), \mu(n) \rangle_b - \langle \mu(n), \mu(n') \rangle_b \geqslant \mu(n)^\top \mu(n) - \mu(n)^\top \mu(n') - 2C'\sqrt{N} \cdot \zeta \tag{25a}$$

$$\geqslant \frac{1}{2N^4} - 2C'\sqrt{N} \cdot 2^{-b-1} \tag{25b}$$

$$= \frac{1}{2N^4} - C'\sqrt{N} \cdot 2^{-b}. \tag{25c}$$

Taking $C = C'$ and noting that $b \geqslant 5\log N$ implies $C\sqrt{N} \cdot 2^{-b} \leqslant C\sqrt{N} \cdot N^{-5} = CN^{-9/2} \leqslant \frac{1}{4N^4}$ for sufficiently large $N$, we obtain

$$\langle \mu(n), \mu(n) \rangle_b - \langle \mu(n), \mu(n') \rangle_b \geqslant \frac{1}{4N^4}. \tag{26}$$

∎

The following follows readily.

**Corollary B.1.** *Let $N \in \mathbb{N}$, $n, n' \in [N]$, $b \geqslant 6\log N$, and let $n_{tgt} \in [N]$ be a target position. Define $\boldsymbol{q}, \boldsymbol{k}_{n'} \in \mathbb{R}^3$ by*

$$\boldsymbol{q}_n \overset{\text{def}}{=} \begin{pmatrix} N^5 \sqrt{\frac{1}{n_{tgt}}} \\ N^5 \sqrt{1 - \frac{1}{n_{tgt}}} \\ -N^5 \end{pmatrix} \quad and \quad \boldsymbol{k}_{n'} \overset{\text{def}}{=} \begin{pmatrix} \sqrt{\frac{1}{n'}} \\ \sqrt{1 - \frac{1}{n'}} \\ 1 \end{pmatrix}, \tag{27}$$

*where all operations are computed in b-bit fixed-point arithmetic. Then, for all $n' \in [N]$:*

$$\begin{cases} |\langle \boldsymbol{q}_n, \boldsymbol{k}_{n'} \rangle_b| \leqslant CN^5 \sqrt{N} \cdot 2^{-b-1} \leqslant CN^{-\frac{1}{2}} & \textbf{\textit{if }} n' = n_{tgt} \\ \langle \boldsymbol{q}_n, \boldsymbol{k}_{n'} \rangle_b \leqslant -C'N & \textbf{\textit{otherwise}} \end{cases} \tag{28}$$

*for some absolute constants $C, C' > 0$ and sufficiently large $N$.*

*Proof.* Write $\boldsymbol{u}_n \overset{\text{def}}{=} \mu(n) = (u_{n',1}, u_{n',2}) \in \mathbb{R}^2$ for the unit-length PE of Lem. B.4 and $\boldsymbol{u}'_n = (u'_{n',1}, u'_{n',2}) \in \mathbb{R}^2$ for its fixed-point realization (so $\boldsymbol{k}_{n'} = (u'_{n',1}, u'_{n',2}, 1)$ and $\boldsymbol{q}_n = (N^5 u'_{n_{tgt},1}, N^5 u'_{n_{tgt},2}, -N^5)$). By construction, $\langle \boldsymbol{q}_n, \boldsymbol{k}_{n'} \rangle_b = N^5 \cdot \left( \langle \boldsymbol{u}'_{n_{tgt}}, \boldsymbol{u}'_{n'} \rangle_b - 1 \right)$ up to the final rounding step.

For $n' = n_{tgt}$, Lem. B.4's proof gives $\left| \langle \boldsymbol{u}'_{n_{tgt}}, \boldsymbol{u}'_{n_{tgt}} \rangle_b - 1 \right| \leqslant C_0 \sqrt{N} \cdot 2^{-b-1}$ for an absolute constant $C_0 > 0$ (the fixed-point error accumulated in the inner product). Scaling by $N^5$ and absorbing the outer rounding step into the constant gives $\left| \langle \boldsymbol{q}_n, \boldsymbol{k}_{n_{tgt}} \rangle_b \right| \leqslant CN^5 \sqrt{N} \cdot 2^{-b-1}$ for some absolute $C > 0$. For $b \geqslant 6 \log N$, this is at most $CN^{11/2-6} = CN^{-1/2}$.

For $n' \neq n_{tgt}$, Lem. B.4 gives $\langle \boldsymbol{u}'_{n_{tgt}}, \boldsymbol{u}'_{n_{tgt}} \rangle_b - \langle \boldsymbol{u}'_{n_{tgt}}, \boldsymbol{u}'_{n'} \rangle_b \geqslant \frac{1}{4N^4}$, and by above $\langle \boldsymbol{u}'_{n_{tgt}}, \boldsymbol{u}'_{n_{tgt}} \rangle_b \leqslant 1 + C_0 \sqrt{N} \cdot 2^{-b-1}$. Combining these using $b \geqslant 6 \log N$, $\langle \boldsymbol{u}'_{n_{tgt}}, \boldsymbol{u}'_{n'} \rangle_b - 1 \leqslant -\frac{1}{4N^4} + C_0 \sqrt{N} \cdot 2^{-b-1} \leqslant -\frac{1}{8N^4}$ for sufficiently large $N$. Multiplying by $N^5$ yields $\langle \boldsymbol{q}_n, \boldsymbol{k}_{n'} \rangle_b \leqslant -\frac{N}{8}$, which gives the claim with $C' = \frac{1}{8}$ (absorbing the final rounding into $C'$). ∎

The following lemma shows that the PEs defined above can be used to focus on individual positions of interest.

**Lemma B.5.** *Let $\boldsymbol{q}_n$ and $\boldsymbol{k}_{n'}$ be defined either as in Lem. B.3 or Cor. B.1 with target index $n_{tgt} \in [N]$ (with corresponding precision requirements). Let $\boldsymbol{s} = (\langle \boldsymbol{q}_n, \boldsymbol{k}_{n'} \rangle_b)_{n' \in [N]}$ for some $N \in \mathbb{N}$ and $n \in [N]$. Then,*

$$\text{softmax}(s)_{n'} = \begin{cases} 1 & \textbf{\textit{if }} n' = n_{tgt} \\ 0 & \textbf{\textit{otherwise}} \end{cases}. \tag{29}$$

*Proof.* The case of Lem. B.3 follows directly from $\langle \boldsymbol{q}_n, \boldsymbol{k}_{n'} \rangle_b \leqslant -2B_{\mathbb{F}}/m$ for $n' \neq n_{tgt}$.

To ensure no attention is placed to positions $n' \neq n_{tgt}$ in Cor. B.1, it suffices to ensure $\langle \boldsymbol{q}_n, \boldsymbol{k}_{n'} \rangle_b \leqslant -\log 2 \, (b+1)$ (cf. Lem. B.2) while $\langle \boldsymbol{q}_n, \boldsymbol{k}_{n_{tgt}} \rangle_b > -\log 2 \, (b+1)$. Substituting the bounds from Cor. B.1, the off-target case satisfies $\langle \boldsymbol{q}_n, \boldsymbol{k}_{n'} \rangle_b \leqslant -C'N$, which dominates $-\log 2 \, (b+1) = -\mathcal{O}(\log N)$ for sufficiently large $N$; the on-target case satisfies $\left| \langle \boldsymbol{q}_n, \boldsymbol{k}_{n_{tgt}} \rangle_b \right| \leqslant CN^{-1/2}$, which is greater than $-\log 2 \, (b+1)$ for sufficiently large $N$. Thus, with logarithmic precision of at least $6 \log N$, softmax attention exclusively attends to individual positions with large attention scores. ∎

### B.2. Useful Attention Patterns

**Lemma B.6** (Detecting a symbol occurrence). *There exists a L-uniform $\text{LPT}^0_{c,1}$ family of transformers $\{\mathcal{T}_N\}_{N \in \mathbb{N}}$ such that, for any $N \in \mathbb{N}$, on input $\boldsymbol{w} \in \Sigma^*$ of length $N$ and $w \in \Sigma$, $\mathcal{T}_N$'s residual stream at position $N$ contains the entry $\mathbb{1}\{w \in \boldsymbol{w}\}$.*

*Proof sketch.* Note that $\mathcal{T}_N$ cannot use the commonly-used *exact* uniform attention over all symbols to detect $\mathbb{1}\{w \in \boldsymbol{w}\}$ due to constant precision. Nevertheless, constant-precision rounded uniform attention suffices. By attending to all symbols in the string with weight 1, the denominator of the attention scores is at most $B_{\mathbb{F}}$. Using one-hot encodings of symbols $w_n$ as the attention values $\boldsymbol{v}_n$, it is easy to see that the final contextual representation at the final position will have a positive value at the entry corresponding to $w$ if and only if $w \in \boldsymbol{w}$, since $c/B_{\mathbb{F}} > 0$ for any $c \geqslant 1$. This condition can be checked by the MLP applied after the attention aggregation operation. This construction requires parameters that do not change with input length (since the size of the one-hot encodings is constant), and is thus logspace computable. ∎

**Lemma B.7** (Reading binary positional encodings into the residual stream). *There exists a L-uniform $\text{LPT}^1_{c,1}$ family of transformers $\{\mathcal{T}_N\}_{N \in \mathbb{N}}$ such that, on input $\& \, B(n)$, $\mathcal{T}_N$'s residual stream at position $\lceil \log N \rceil + 1$ contains the value $B(n)$ in a designated $\lceil \log N \rceil$-dimensional sub-block for any $N \in \mathbb{N}$ and $n \in [N]$.*

*Proof sketch.* The transformer $\mathcal{T}_N$ has to convert the binary representation $B(n)$ of $n$ contained across $\lceil \log N \rceil$ positions in the input string into a single $\lceil \log N \rceil$-dimensional binary vector in the residual stream. The construction uses a fixed

single-layer block that is looped $\lceil \log N \rceil$ times—i.e., constant depth and $\mathcal{O}(\log N)$ loop iterations of a single block, as required by $\mathtt{LPT}^1_{\mathrm{c,1}}$. We index loop iterations by a timestep $t \in \{1, \ldots, \lceil \log N \rceil\}$.

1. At timestep $t = 1$ (the first application of the looped block), each symbol $w_{n'} \in \{0, 1\}$ checks if it is immediately preceded by the & symbol, which denotes the beginning of the pointer in the string. If it is, $w_{n'}$ stores $\boldsymbol{e}_1$ and $\boldsymbol{d}_1 \overset{\text{def}}{=} w_{n'}\boldsymbol{e}_1$ in designated parts of its residual stream. Here, $\boldsymbol{e}_1$ is the first unit vector of $\mathbb{R}^{\lceil \log N \rceil}$.

2. At each subsequent timestep $t \in \{2, \ldots, \lceil \log N \rceil\}$ (i.e., the $t$-th application of the same single-layer block), each symbol $w_{n'}$ checks if the entry $\boldsymbol{e}_{t-1}$ has already been written to the designated space of the previous symbol's residual stream. If it has, $w_{n'}$ copies and shifts $\boldsymbol{e}_{t-1}$ into $\boldsymbol{e}_t$, and stores $\boldsymbol{e}_t$ and $\boldsymbol{d}_t \overset{\text{def}}{=} \boldsymbol{d}_{t-1} + w_{n'}\boldsymbol{e}_t$ in designated parts of its residual stream.

After $\lceil \log N \rceil$ timesteps (i.e., $\lceil \log N \rceil$ applications of the looped block), the residual stream at position $\lceil \log N \rceil + 1$ thus contains $\mathtt{B}(n)$.

This construction only requires PEs that contain $\mathtt{B}(n)$ and $\mathtt{B}(N - 1)$, which are logarithmically computable. Moreover, the parameters of the transformer $\mathcal{T}_N$ only change with $N$ in terms of the size of the matrices, while their structure remains the same—they either project onto specific coordinates (whose indices can be computed with counters in a logspace Turing machine) or shift the coordinates of vectors, which can also be done in logspace. Thus, the family $\{\mathcal{T}_N\}_{N \in \mathbb{N}}$ is in L-uniform $\mathtt{LPT}^1_{\mathrm{c,1}}$. ∎

### B.3. Layer Normalization

For compactness, our constructions ignore layer normalization (apart from those extending Merrill & Sabharwal's (2025a) constructions, which rely on layer normalization for implementing the layer hash norm). However, it is not difficult to account for it. For upper bounds, it suffices to see that, in the case of *constant precision* $b = \Theta(1)$, any position-wise operation (such as layer normalization) can be implemented by a finite lookup table, which can be hardcoded into the MLPs of the transformer. In the case of growing precision, all layer normalization operations (addition, division, square root, etc.) can be implemented in L-uniform $\mathtt{TC}^0$ (Chiang, 2025), ensuring that the same upper bounds still hold. For lower bounds, we can use the same construction as Li et al., 2024b, §F.1, which minimally changes the constructed transformers to ensure that the layer normalization operation does not affect the computations.

Whenever we do refer to layer normalization, we use the standard numerically-stable variant of layer normalization

$$\boldsymbol{\nu}(\boldsymbol{x}) \overset{\text{def}}{=} \frac{\boldsymbol{x} - \mu(\boldsymbol{x})}{\sqrt{\sigma^2(\boldsymbol{x}) + \varepsilon}}, \tag{30}$$

where $\mu(\boldsymbol{x})$ and $\sigma^2(\boldsymbol{x})$ denote the empirical mean and variance of the coordinates of $\boldsymbol{x}$, and $\varepsilon \geqslant 2^{-b}$ is a fixed stabilizer.

## C. Proofs

### C.1. Proofs of the Results on the Relationship between AHATs and SMATs

**Lemma 3.1.** *Let* $\{\mathcal{T}\}_{N \in \mathbb{N}}$ *be a logarithmic-precision* L-*uniform* AHAT *family. There exists a logarithmic-precision* L-*uniform* SMAT *family* $\{\mathcal{T}'\}_{N \in \mathbb{N}}$ *such that for any input* $\boldsymbol{w} \in \Sigma^N$ *and* $N \in \mathbb{N}$*, the outputs of* $\mathcal{T}_N$ *and* $\mathcal{T}'_N$ *match.*

*Proof.* The construction is to keep all parameters of $\mathcal{T}_N$ fixed and replace every average-hard attention layer by a temperature-scaled softmax attention layer with a suitably small temperature $\tau(N)$. We first establish, in $\mathbb{R}$-valued arithmetic, a per-layer error bound between $\mathcal{T}_N$ and its softmax counterpart in terms of the attention gap and the temperature. We then choose the temperature small enough that the per-coordinate error is below the fixed-point rounding margin, so that the rounding step in §A.3 collapses the two models onto identical outputs. For brevity, write $b \overset{\text{def}}{=} b(N) = \Theta(\log N)$, $\mathbb{F} \overset{\text{def}}{=} \mathbb{F}_b$, and let $L$ denote the number of layers of $\mathcal{T}_N$.

**Step 1: Gap and activation bounds.** Following Yang et al. (2026a, Def. 5), for an attention layer with scores $s_{n,1}, \ldots, s_{n,N} \in \mathbb{F}$ at position $n$ with maximum $s_{n,\max} \overset{\text{def}}{=} \max_{n'} s_{n,n'}$, the layer has **gap** $\gamma(N)$ at position $n$ if $s_{n,n'} \leqslant s_{n,\max} - \gamma(N)$ for every $n'$ with $s_{n,n'} < s_{n,\max}$. The gap of the layer is the largest such $\gamma(N)$ that works

for all $n$ and all length-$N$ inputs, and the gap of the transformer is the minimum over its $L$ layers. Because every score is an element of $\mathbb{F}$ and any two distinct elements of $\mathbb{F}$ differ by at least $2^{-b}$, we have

$$\gamma(N) \; \geqslant \; 2^{-b} \; = \; \Omega(N^{-c}) \tag{31}$$

for some constant $c$ depending only on the constant hidden in $b = \Theta(\log N)$. Similarly, every activation $\boldsymbol{h}_n^{(l)} \in \mathbb{F}^D$ has every coordinate in $[-B_{\mathbb{F}}, B_{\mathbb{F}}]$ with $B_{\mathbb{F}} = 2^b - 2^{-b}$, so the analogue of Yang et al.'s (2026a) $x_{\max}(N)$ (maximum value in any entry of the residual stream of length-$N$ strings) satisfies

$$x_{\max}(N) \; \leqslant \; B_{\mathbb{F}} \; = \; \mathcal{O}(N^c). \tag{32}$$

The same bound applies to the parameter matrices $\boldsymbol{W}_Q^{(l)}, \boldsymbol{W}_K^{(l)}, \boldsymbol{W}_V^{(l)}$ and the MLP parameters, since they too are in $\mathbb{F}$.

Let $\mathcal{T}_N'$ be the candidate softmax model: It has the same parameters as $\mathcal{T}_N$ except that every attention layer uses $\mathrm{softmax}_\tau$ with temperature $\tau(N)$ in place of $\mathrm{ahardmax}$. Both models operate under the same fixed-point semantics from §A.3, so every layer's output is rounded coordinate-wise to $\mathbb{F}^D$.

**Step 2: Per-layer error in $\mathbb{R}$-valued arithmetic.** Fix any layer $l \in \{1, \ldots, L\}$ and any input $\boldsymbol{H}^{(l-1)} \in \mathbb{F}^{N \times D}$ shared by both models. Let $\boldsymbol{h}_n^{(l)}$ and $\widetilde{\boldsymbol{h}}_n^{(l)}$ denote the layer-$l$ outputs of $\mathcal{T}_N$ and $\mathcal{T}_N'$ respectively when both are computed in idealized $\mathbb{R}$-valued arithmetic on this shared input (i.e., without the rounding step). By Yang et al.'s (2026a, Lem. 25) per-layer bound (specialized to zero input error, since the two models share their layer input), there is a constant $K_1 \geqslant 1$ depending polynomially on $D$ and the maximum parameter magnitude such that, for every position $n$,

$$\left\| \boldsymbol{h}_n^{(l)} - \widetilde{\boldsymbol{h}}_n^{(l)} \right\|_1 \; \leqslant \; K_1 \, x_{\max}(N) \, N \, e^{-\gamma(N)/\tau(N)}. \tag{33}$$

**Step 3: Choosing the temperature so a single layer's error rounds away.** We want the implemented attention sub-layer's output to lie strictly below half the spacing $2^{-b}$ between adjacent elements of $\mathbb{F}$ from $\mathcal{T}_N$'s hardmax-attention output; this guarantees that every coordinate of $\widetilde{\boldsymbol{h}}_n^{(l)}$ lies strictly closer to the corresponding coordinate of $\boldsymbol{h}_n^{(l)} \in \mathbb{F}$ than to any other element of $\mathbb{F}$, and therefore rounds to that same element. Let $K_2 \geqslant 1$ be a constant bound on the per-layer Lipschitz constant of the remaining sub-layers (value projection, residual sum, MLP, Lipschitz layer normalization with stabilizer $\varepsilon$; cf. §B.3) so that they amplify $\mathbb{R}$-arithmetic errors by at most a factor of $K_2$. Solving $K_1 K_2 \, x_{\max}(N) \, N \, e^{-\gamma(N)/\tau(N)} < 2^{-(b+2)}$ for $\tau(N)$—tightened to $2^{-(b+2)}$ so that the Step 4 mixed-precision rounding (which contributes at most $2^{-(b+2)}$ per coordinate; cf. Step 4(ii)) and the Step 5 sub-layer amplification still leave a margin strictly below $2^{-(b+1)}$—yields the condition

$$\frac{1}{\tau(N)} \; > \; \frac{1}{\gamma(N)} \log\bigl(2^{b+2} K_1 K_2 \, x_{\max}(N) \, N\bigr). \tag{34}$$

Plugging in $b = \Theta(\log N)$, $\gamma(N) = \Omega(N^{-c})$ from Eq. (31), and $x_{\max}(N) = \mathcal{O}(N^c)$ from Eq. (32) gives

$$\frac{1}{\tau(N)} \; \in \; \mathcal{O}(N^c \log N) \; \subseteq \; \mathcal{O}(N^{c+1}), \tag{35}$$

so $1/\tau(N)$ is a polynomial in $N$ and is logspace-computable. We fix $\tau(N) \in \mathbb{F}$ to be any value satisfying Eq. (34).

**Step 4: $\mathcal{T}_N'$ as a simulator of $\mathcal{T}_N$.** We define $\mathcal{T}_N'$ to have the same parameters as $\mathcal{T}_N$ (same $\boldsymbol{W}_Q, \boldsymbol{W}_K, \boldsymbol{W}_V$, same MLP, same layer normalization, same PEs), at the same residual-stream precision $b$, with each attention layer using $\mathrm{softmax}_\tau$ at the temperature $\tau(N)$ from Step 3. Since $\tau(N)$ from Step 3 satisfies $1/\tau(N) \in \mathcal{O}(N^{c+1})$, choosing it to be a negative power of two—permitted since Eq. (34) only constrains $1/\tau(N)$ from below—makes $\tau(N)$ logspace-computable from $1^N$.

However, the unnormalized attention scores $s_{n,n'} = \boldsymbol{q}_n^\top \boldsymbol{k}_{n'} \in \mathbb{F}_b$ are scaled by $1/\tau(N)$ before being passed through the softmax, meaning that the scaled scores $s_{n,n'}/\tau(N)$ can be polynomially larger in magnitude than $B_{\mathbb{F}}$, and thus not representable in $\mathbb{F}_b$. We handle this with mixed-precision (cf. §A.4)—narrowed to a single component: The attention sub-layer computes the scaled scores and their softmax at a higher precision $b' \stackrel{\mathrm{def}}{=} \kappa \cdot b$ for a constant $\kappa > 1$, after which the resulting attention weights and the value-weighted sum are clamped back to $\mathbb{F}_b$ when written to the residual stream. All other components of the transformer remain at the original precision $b$: The mixed-precision is used only for the internal score-and-softmax computation. We pick $\kappa$ large enough that

(i) the scaled scores $s_{n,n'}/\tau(N)$ are exactly representable in $\mathbb{F}_{b'}$, and

(ii) the accumulated rounding error in computing the softmax and the value-weighted sum at precision $b'$ is at most $2^{-(b+2)}$ per output coordinate.

Using $1/\tau(N) \in \mathcal{O}(N^{c+1})$, $B_{\mathbb{F}} \in \mathcal{O}(N^c)$, and $D \leqslant \texttt{poly}(N)$, any constant $\kappa$ with $\kappa b \geqslant b + \log_2(ND/\tau(N)) + \Theta(1)$ suffices for both (i) and (ii); such a $\kappa$ depends only on the constant hidden in $b = \Theta(\log N)$.

**Step 5: Layer-by-layer agreement in $\mathbb{F}_b$ and L-uniformity of $\mathcal{T}'_N$.** Bound (ii) above plays the same role as Step 2's per-layer error bound, but at the level of the implemented attention sub-layer rather than its idealized $\mathbb{R}$-arithmetic counterpart: At every layer $l$, the clamped output of $\mathcal{T}'_N$'s attention sub-layer matches $\mathcal{T}_N$'s hardmax-attention output to within $2^{-(b+2)}$ in $\mathbb{R}$. The remaining components amplify this by at most a factor of $K_2$, which is absorbed by the $K_2$ factor already in Eq. (34) (Step 3); the resulting total per-layer error remains strictly below the rounding threshold $2^{-(b+1)}$. Standard induction on $l$ then yields

$$\boldsymbol{H}'^{(l)} = \boldsymbol{H}^{(l)} \qquad \text{in } \mathbb{F}_b^{N \times D}, \tag{36}$$

for every $l$: The base case $l = 0$ holds because both models share the same embedding and PEs, and the inductive step uses the per-layer bound above with the shared input $\boldsymbol{H}^{(l-1)}$ given by IH to conclude that the $\mathbb{F}_b$ rounding of $\mathcal{T}'_N$'s layer-$l$ output collapses onto $\mathcal{T}_N$'s layer-$l$ output. In particular, the final outputs of $\mathcal{T}'_N$ and $\mathcal{T}_N$ coincide in $\mathbb{F}_b^{N \times D}$.

For L-uniformity of $\mathcal{T}'_N$: By construction, $\mathcal{T}'_N$ has the same parameters as $\mathcal{T}_N$ except for the additional temperature $\tau(N)$, which is logspace-computable from $1^N$ (a negative power of two with $\log_2(1/\tau(N)) = \mathcal{O}(\log N)$ bits). The construction machine $\mathcal{M}_1$ of Def. A.5 is therefore unchanged for the weight matrices and the MLP, and emits one additional logspace-computable quantity (the temperature); the PE machine $\mathcal{M}_2$ is unchanged. Hence $\mathcal{T}'_N$ is an L-uniform logarithmic-precision $\tau$-SMAT family whose outputs coincide with $\mathcal{T}_N$'s on every input by Eq. (36), as the lemma claims. ∎

## C.2. Proofs of the Constant-depth Results

**Lemma 4.1.** L-*uniform* $\mathtt{TC}^0 \subseteq$ L-*uniform* $\mathtt{LPT}^0_{1,c}$.

*Proof.* The proof follows London & Kanade (2025, Thm. C.7), which constructs an L-uniform $\mathtt{LPT}^0_{1,1}$ SMAT family simulating a given L-uniform $\mathtt{TC}^0$ circuit family $\{C_N\}_{N \in \mathbb{N}}$. Our construction reuses theirs verbatim except for the PE: Wherever the original construction stores or queries a binary-valued (i.e., constant-precision) pointer of width $\lceil \log N \rceil$ (containing $\mathsf{B}(n)$), we substitute the two-dimensional unit-length PE $\mu$ of Eq. (19). We describe the substitution concretely below.

**Symbol types and original residual-stream layout.** Following London & Kanade (2025, §C.1), the input to the transformer is a residual stream with entries of three types: Input symbols $\mathsf{Inp}(n)$ (entered from the input string), argument symbols $\mathsf{Arg}(n_g, n_a)$ (encoding that gate $n_g$ takes the value at position $n_a$ as one of its arguments; entered from the PEs), and gate symbols $\mathsf{Type}(n_g)$ (encoding the type and threshold of gate $n_g$; also entered from the PEs). Here, $n, n_g, n_a \in [N]$ are positions in the padded input sequence: $n_g$ is the position of the gate symbol itself, and $n_a$ is the position of the source symbol (an $\mathsf{Inp}(\cdot)$ or a previously-placed $\mathsf{Type}(\cdot)$) feeding into it. In their construction, each symbol's PE has the form

$$\phi_{\text{bin}}(\cdot) = (c_1, c_2, c_3, \boldsymbol{k}_{n_{\text{tgt}_1}}, \boldsymbol{q}_{n_{\text{tgt}_1}}, \boldsymbol{k}_{n_{\text{tgt}_2}}, \boldsymbol{q}_{n_{\text{tgt}_2}}), \tag{37}$$

where $c_1, c_2, c_3 \in \{0, 1\}$ are constant-size flags, and each $(\boldsymbol{k}_\bullet, \boldsymbol{q}_\bullet)$ pair is the signed-binary key–query construction of Lem. B.3, i.e., $\boldsymbol{k}_{n_\bullet} = \mathsf{B}^\pm(n_\bullet) \frown \boldsymbol{1}_{\lceil \log N \rceil}$ and $\boldsymbol{q}_{n_\bullet} = B_{\mathbb{F}} \cdot \mathsf{B}^\pm(n_\bullet) \frown (-\boldsymbol{1}_{\lceil \log N \rceil})$. The two stored positions $n_{\text{tgt}_1}, n_{\text{tgt}_2} \in [N]$ are the two attention targets a symbol must address—one per attention layer of the two-layer-per-circuit-layer construction, each an instance of the target $n_{\text{tgt}}$ from Lem. B.4. Specifically, in the first layer, an $\mathsf{Arg}(n_g, n_a)$ symbol *reads* the value at the source position $n_{\text{tgt}_1} = n_a$ into its own slot, and in the second layer, a $\mathsf{Type}(n_g)$ symbol *collects* and *computes* the value of the gate at its own index $n_{\text{tgt}_2} = n_g$. Each binary key/query block occupies $2\lceil \log N \rceil$ coordinates, so the residual stream has width $\Theta(\log N)$.

**Constant-width substitution.** We replace Eq. (37) by

$$\phi_{\text{unit}}(\cdot) = (c_1, c_2, c_3, \mu(n_{\text{tgt}_1}), \mu(n_{\text{tgt}_2})), \tag{38}$$

so each of the two log-width $(\boldsymbol{k}_\bullet, \boldsymbol{q}_\bullet)$ blocks in Eq. (37) collapses to a single two-dimensional unit-length PE. The total PE width is $3 + 2 + 2 = 7$, i.e., constant in $N$. The machine $\mathcal{M}_1$ from Def. A.5 that generates the transformer's parameters is

identical to London et al.'s. The machine $\mathcal{M}_2$ that computes the PEs, in contrast, writes $\mu(n_\bullet)$ whenever their construction would write a $(\boldsymbol{k}_{n_\bullet}, \boldsymbol{q}_{n_\bullet})$ block into the PE; this PE is logspace-computable from $n_\bullet$ and $N$ by Lem. B.4.

**Attention with unit-length PEs.**    The substitution preserves the attention pattern of London & Kanade (2025). Each attention layer must focus exclusively on a single target, namely $n_{\text{tgt}_1}$ in the first layer and $n_{\text{tgt}_2}$ in the second. Cor. B.1 provides the query–key construction: Scaling yields a query $\boldsymbol{q}_n \stackrel{\text{def}}{=} {}^{B_\mathbb{F}}/m \cdot \mu(n_{\text{tgt}})$ and key $\boldsymbol{k}_{n'} \stackrel{\text{def}}{=} \mu(n')$ such that the inner-product gap between $n' = n_{\text{tgt}}$ and $n' \neq n_{\text{tgt}}$ exceeds the saturation threshold required by Lem. B.5; we instantiate this with $n_{\text{tgt}} = n_{\text{tgt}_1}$ in the first layer and $n_{\text{tgt}} = n_{\text{tgt}_2}$ in the second. Both targets are already coordinates of the symbol's PE (cf. Eq. (38)), so the query and key matrices are simple projections—with the ${}^{B_\mathbb{F}}/m$ scaling absorbed into them, as in the proof of Lem. 3.1—onto those two coordinates. Concretely, in the first attention layer the query of each symbol projects out $\mu(n_{\text{tgt}_1})$ from its PE while every symbol's key projects out $\mu(n_{\text{tgt}_2})$; since each non-pause symbol's $n_{\text{tgt}_2}$ equals its own position in London & Kanade's (2025) encoding, an argument symbol with $n_{\text{tgt}_1} = n_a$ attends exactly to the source at position $n_a$. The second layer swaps the roles—query from $\mu(n_{\text{tgt}_2})$, key from $\mu(n_{\text{tgt}_1})$—so that every argument symbol with $n_{\text{tgt}_1} = n_g$ is attended to by the gate at position $n_g$, matching London et al.'s second-layer pattern. By Lem. B.5, the resulting post-softmax weights are exactly 1 on the target position and 0 elsewhere in $\mathbb{F}_b$ arithmetic. The value matrix, MLP, and the rest of the residual-stream bookkeeping are inherited verbatim from London & Kanade (2025, Thm. C.7), since they act position-wise on the constant-size flags $c_1, c_2, c_3$ and the (already constant-size) gate-value coordinate, none of which touch the PE block.

**Residual-stream contents remain constant-width.**    Beyond the PEs in Eq. (38), the residual stream stores only a constant number of additional coordinates: The computed gate value (one coordinate with value in $\mathbb{F}_b$) and the temporary scratch coordinates used by the MLP to recombine flags into thresholds—all inherited verbatim from London & Kanade (2025) and already constant-size there. What changes is only the *PE block*: From $\Theta(\log N)$ coordinates of binary key/query pairs to 4 coordinates of unit-length PEs. The overall width is therefore $\mathcal{O}(1)$.

Finally, we note that London & Kanade's (2025) construction effectively constructs an AHAT (an SMAT in which the non-max values are small enough to map to 0 via the softmax; cf. Lem. B.5), meaning that this constructions show the same relationship for AHATs as well. ∎

**Lemma 4.2.** L-*uniform* $\text{LPT}^0_{\text{c},\text{p}} \subseteq$ L-*uniform* $\text{AC}^0$ *and* L-*uniform* $\text{LPT}^0_{\text{p},\text{p}} \subseteq$ L-*uniform* $\text{TC}^0$.

*Proof.*  The proof follows London & Kanade (2025, Thms. C.5 and C.11), who handle $\text{LPT}^0_{\text{c},1}$ and $\text{LPT}^0_{1,1}$, respectively, by decomposing each transformer layer into components that are individually in L-uniform $\text{AC}^0$ (constant-precision case) or L-uniform $\text{TC}^0$ (log-precision case) and composing a constant number of them. We follow the same decomposition but track how each component scales when both width and precision are polynomial. Fix a L-uniform SMAT family $\{\mathcal{T}_N\}_{N \in \mathbb{N}}$ of constant depth and let $b(N), D(N) \in \texttt{poly}(N)$ denote its precision and width, respectively.

**Positional encodings and per-coordinate parameters.**    Def. A.5 guarantees that PE coordinates and parameter-matrix entries are produced by a logspace TM, given the input length $N$, the relevant position $n \in [N]$, and the coordinate indices. Concatenating all $N \cdot D(N) \cdot b(N) = \texttt{poly}(N)$ output bits gives a logspace-computable function of $1^N$, so Lem. B.1 yields an L-uniform $\text{AC}^0$ sub-circuit of polynomial size and depth 3 that produces every PE and parameter bit consumed by the attention and feedforward components, just as in London & Kanade's (2025) proofs of Thms. C.5 and C.11 use of the same lemma.

**Attention layer.**    Per head and per query position $n$, the layer computes attention scores $s_{n,n'} \stackrel{\text{def}}{=} \langle \boldsymbol{q}_n, \boldsymbol{k}_{n'} \rangle_{b(N)}$ for $n' \in [N]$, normalizes them via softmax, and returns the weighted value sum. Each $s_{n,n'}$ is an inner product of two $D(N)$-dimensional vectors with $b(N)$-bit entries—i.e., a sum of $\texttt{poly}(N)$ products of $\texttt{poly}(N)$-bit numbers; multiplication of two $\texttt{poly}(N)$-bit numbers is in L-uniform $\text{TC}^0$ (Hesse et al., 2002), and iterated addition of $\texttt{poly}(N)$ such products is in L-uniform $\text{TC}^0$ by Hesse et al. (2002, Thm. 2.1). Evaluating $\exp$ on a $\texttt{poly}(N)$-bit fixed-point input to $\texttt{poly}(N)$-bit precision is also in L-uniform $\text{TC}^0$: It is computed by a truncated Taylor series whose terms reduce to iterated multiplication and iterated addition, both of which are in L-uniform $\text{TC}^0$ (Chiang, 2025). Hence the unnormalized weights $\exp(s_{n,n'})$ and their iterated sum (the normalizer) are in L-uniform $\text{TC}^0$ as well. Dividing each $\exp(s_{n,n'})$ by the normalizer and computing the new value involves $\texttt{poly}(N)$ divisions and one more iterated sum per output coordinate, both in L-uniform $\text{TC}^0$. Composing a constant number of L-uniform $\text{TC}^0$ subroutines keeps the layer in L-uniform $\text{TC}^0$. In the constant-precision case the same

decomposition collapses to L-uniform $\mathtt{AC}^0$: Each product becomes a finite lookup table (London & Kanade, 2025, Lem. C.4), exponentiation reduces to a lookup table as well (London & Kanade, 2025, Lem. D.4), and iterated addition of $\mathrm{poly}(N)$ constant-precision numbers is in L-uniform $\mathtt{AC}^0$ by London & Kanade (2025, Thm. C.3).

**Feedforward layer and residual connection.** A feedforward layer applies a position-wise affine map followed by a ReLU nonlinearity. Each output coordinate is a sum of $D(N)$ products of $b(N)$-bit numbers plus a ReLU thresholding—both in L-uniform $\mathtt{TC}^0$ at polynomial precision and in L-uniform $\mathtt{AC}^0$ at constant precision by the same reasoning as above. The residual connection adds two $b(N)$-bit numbers position-wise, which is in L-uniform $\mathtt{AC}^0$ even at polynomial precision (Hesse et al., 2002).

**Composition.** A constant-depth transformer applies a constant number of attention and feedforward layers (each in L-uniform $\mathtt{TC}^0$ for polynomial precision, L-uniform $\mathtt{AC}^0$ for constant precision), so the whole computation lies in the same class. This yields L-uniform $\mathtt{LPT}^0_{\mathsf{c,p}} \subseteq$ L-uniform $\mathtt{AC}^0$ and L-uniform $\mathtt{LPT}^0_{\mathsf{p,p}} \subseteq$ L-uniform $\mathtt{TC}^0$.

**Extension to AHATs.** For AHATs, the softmax-and-value-sum step is replaced by argmax-then-average: For each query position $n$, identify the set of positions in $\mathrm{argmax}_{n'}\, s_{n,n'}$ and return the average of their values. Computing the maximum of $N$ polynomial-precision integers and counting the argmax positions are both in FO-uniform $\mathtt{AC}^0 \subseteq$ L-uniform $\mathtt{TC}^0$ (Chiang, 2025, Thm. 2), and the resulting average is again iterated addition followed by one division—in L-uniform $\mathtt{TC}^0$ at polynomial precision and in L-uniform $\mathtt{AC}^0$ at constant precision. Hence both inclusions extend to AHATs. ∎

### C.3. Proofs of Polylogarithmic-depth Results

**Lemma 5.1** (Lem. 9). *Let $m(N)$, $d(N)$, and $r(N)$ be functions at most polynomial in $N$ and logspace-computable given $1^N$ such that $r(N) \cdot d(N) \geqslant 1$. Let $f \colon \{0,1\}^{m(N)} \to \{0,1\}^{m(N)}$ be computed by an L-uniform polynomial-size circuit family of depth $\mathcal{O}(d(N))$ with AND, OR, NOT gates (resp. additionally with THR gates). Then $f^{r(N)}$ is computed by an FO-uniform polynomial-size circuit family of depth $\mathcal{O}(r(N)\, d(N))$ with the same gate set.*

*Proof.* We follow the strategy of Merrill & Sabharwal (2025a, Lem. 9), who cover FO-uniform input families. In contrast, we assume an L-uniform circuit family for $f$—so we first construct an L-uniform iterated circuit family for $f^{r(N)}$ by a logspace stitching argument, and then invoke uniformity collapse to strengthen to FO-uniform.

**Step 1: an L-uniform iterated family.** Fix the input length $N$. Let $\mathcal{C}^f = \{C_N^f\}_{N=0}^{\infty}$ be the assumed polynomial-size L-uniform circuit family for $f$, and let $M_f$ be a logspace Turing machine that, given $1^N$ and a gate or edge query, decides membership in the connection language (cf. §A.2) of $C_N^f$. Since $C_N^f$ has polynomially many gates, we may pad $C_N^f$ with dummy gates (handled by a fixed default response from $M_f$) so that it has exactly $\sigma \overset{\text{def}}{=} 2^{s(N)}$ gates for some integer $s(N) = \mathcal{O}(\log N)$; this preserves L-uniformity and at most doubles the circuit size. We further assume, without loss of generality, that the $m(N)$ input gates of $C_N^f$ occupy indices $0, \ldots, m(N) - 1$ and the $m(N)$ output gates occupy indices $\sigma - m(N), \ldots, \sigma - 1$; if they do not, we prepend a layer of $m(N)$ identity gates at indices $0, \ldots, m(N) - 1$ wired to the original input gates and append a layer of $m(N)$ identity gates at indices $\sigma - m(N), \ldots, \sigma - 1$ fed by the original output gates. This adds 2 to the depth and $2m(N)$ to the size, and preserves L-uniformity since the rewiring can be carried out by $M_f$ in logspace.

We construct the circuit $C_N$ as $r(N)$ copies of $C_N^f$ stacked sequentially, with the output gates of each copy wired to the input positions of the next. Indices in $C_N$ range over $\{0, 1, \ldots, r(N) \cdot \sigma - 1\}$; we decompose any such index $i$ uniquely as $i = q_i \cdot \sigma + i'$ with the block index $q_i \in \{0, \ldots, r(N) - 1\}$ and the within-block index $i' \in \{0, \ldots, \sigma - 1\}$. Both components are computable from the $\mathcal{O}(\log N)$-bit representation of $i$ in logspace.

The connection language of $C_N$ is then decided by a logspace machine $M$ that, on input $1^N$ and a query about $C_N$, proceeds as follows:

*(a)* On a **gate query** at index $i$, $M$ rejects if $i \geqslant r(N) \cdot \sigma$; otherwise, it computes $(q_i, i')$ and queries $M_f$ for the gate type of index $i'$ in $C_N^f$. If $M_f$ reports an input gate and $q_i \geqslant 1$, $M$ overrides the response to an identity gate—so that block $q_i$'s "input" positions become internal pass-through gates fed by block $q_i - 1$—and otherwise it returns $M_f$'s response unchanged.

*(b)* On an edge query for the pair $(i, j)$, $M$ computes $(q_i, i')$ and $(q_j, j')$ and proceeds as follows. If $q_i = q_j$, $M$ forwards the edge query $(i', j')$ to $M_f$, reproducing the intra-block edges of $C_N^f$ inside every block. If $q_j = q_i + 1$, $i' \geqslant \sigma - m(N)$, $j' < m(N)$, and $i' - j' = \sigma - m(N)$—equivalently, the $k$-th output gate of block $q_i$ is paired with the $k$-th input position of block $q_i + 1$ for the same $k$—$M$ returns 1, wiring exactly the intended $m(N)$ edges per block boundary. All remaining edge queries return 0.

Every test performed by $M$ is a comparison or addition on $\mathcal{O}(\log N)$-bit numbers, and answering queries about $C_N^f$ is delegated to $M_f$; hence $M$ runs in logspace. By construction, $C_N$ has size $r(N) \cdot \sigma = \texttt{poly}(N)$, depth $\mathcal{O}(d(N) \cdot r(N))$, and computes $f^{r(N)}$. Since $r(N) \cdot d(N)$ is at most polynomial, $\mathcal{C}$ is an L-uniform polynomial-size circuit family of depth $\mathcal{O}(r(N) \, d(N))$ over the same gate set as $\mathcal{C}^f$.

**Step 2: upgrading L-uniformity to FO-uniformity.** Since $r(N) \cdot d(N) \geqslant 1$ whenever $r(N), d(N) \geqslant 1$, the *uniformity collapse* result Merrill & Sabharwal, 2025a, Thm. 3—namely, FO-uniform $\texttt{AC}^d = $ L-uniform $\texttt{AC}^d$ and FO-uniform $\texttt{TC}^d = $ L-uniform $\texttt{TC}^d$ for $d \geqslant 1$—applies, yielding the desired FO-uniform circuit family for $f^{r(N)}$ at depth $d(N) \cdot r(N)$. ∎

**Lemma 5.2** (Lem. 6). *For $d \geqslant 1$:*

$$\text{L-}\textit{uniform } \texttt{LPT}_{\texttt{c,p}}^{\texttt{d}} \subseteq \text{FO-}\textit{uniform } \texttt{AC}^{\texttt{d}} \tag{5a}$$

$$\text{L-}\textit{uniform } \texttt{LPT}_{\texttt{1,p}}^{\texttt{d}} \subseteq \text{FO-}\textit{uniform } \texttt{TC}^{\texttt{d}}. \tag{5b}$$

*Proof.* The argument adapts Merrill & Sabharwal (2025a, Lem. 6), which proves an analogous statement for fully-uniform log-precision AHATs. Despite the different starting point—we begin from L-uniform constant-depth transformers in L-uniform $\texttt{AC}^0$ or L-uniform $\texttt{TC}^0$ instead of fully-uniform log-precision AHATs in FO-uniform $\texttt{TC}^0$—we arrive at the same FO-uniform upper bounds for L-uniform looped transformers. Additionally, the argument now applies to both AHATs and SMATs at constant and growing precision. Fix any family $\{\mathcal{T}_N\}_{N \in \mathbb{N}}$ in L-uniform $\texttt{LPT}_{\texttt{c,p}}^{\texttt{d}}$ or L-uniform $\texttt{LPT}_{\texttt{1,p}}^{\texttt{d}}$ for some $d \geqslant 1$, and let $\mathcal{L}$ be the language it recognizes. Write $\Theta(\log^d N)$ for the loop count and let $\mathcal{T}_N$ have a designated looped block of constant depth (cf. Def. A.6). Following Merrill & Sabharwal (2025a), we partition the $L$ layers $\boldsymbol{\tau}^{(1)}, \ldots, \boldsymbol{\tau}^{(L)}$ of $\mathcal{T}_N$ into three pieces using the loop boundaries $1 \leqslant l_1 \leqslant l_2 \leqslant L$ of Def. A.6: The pre-loop layers $\mathcal{T}_N^A \stackrel{\text{def}}{=} \boldsymbol{\tau}^{(l_1)} \circ \cdots \circ \boldsymbol{\tau}^{(1)}$, the looped constant-depth block $\mathcal{T}_N^B \stackrel{\text{def}}{=} \boldsymbol{\tau}^{(l_2)} \circ \cdots \circ \boldsymbol{\tau}^{(l_1+1)}$ iterated $r(N) \stackrel{\text{def}}{=} \Theta(\log^d N) = T$ times, and the post-loop layers $\mathcal{T}_N^C \stackrel{\text{def}}{=} \boldsymbol{\tau}^{(L)} \circ \cdots \circ \boldsymbol{\tau}^{(l_2+1)}$, so that $\mathcal{T}_N = \mathcal{T}_N^C \circ (\mathcal{T}_N^B)^{r(N)} \circ \mathcal{T}_N^A$.

**Step 1: Each piece is a constant-depth padded transformer.** By construction, each of $\mathcal{T}_N^A$, $\mathcal{T}_N^B$, and $\mathcal{T}_N^C$ is a constant-depth padded transformer with the same width, precision, and attention type as $\mathcal{T}_N$. All three are themselves L-uniform: Their parameter matrices are an L-computable projection (selecting a contiguous range of layers) of those of $\mathcal{T}_N$, and the construction machine $\mathcal{M}_1$ of $\mathcal{T}_N$ only needs to be augmented with logspace-computable counters marking the layer ranges $A, B, C$. The PE machine $\mathcal{M}_2$ is unchanged. Hence each piece lies in L-uniform $\texttt{LPT}_{\texttt{c,p}}^0$ or L-uniform $\texttt{LPT}_{\texttt{p,p}}^0$ depending on the precision regime.

**Step 2: Simulating each piece by a constant-depth circuit.** Applying Lem. 4.2 to $\mathcal{T}_N^A$, $\mathcal{T}_N^B$, and $\mathcal{T}_N^C$ yields L-uniform circuit families $\{C_N^A\}_{N \in \mathbb{N}}$, $\{C_N^B\}_{N \in \mathbb{N}}$, and $\{C_N^C\}_{N \in \mathbb{N}}$ of polynomial size and constant depth that simulate them; The gate set is $\{\texttt{AND}, \texttt{OR}, \texttt{NOT}\}$ in the constant-precision case (so each family is in L-uniform $\texttt{AC}^0$) and $\{\texttt{AND}, \texttt{OR}, \texttt{NOT}, \texttt{THR}\}$ in the growing-precision case (so each is in L-uniform $\texttt{TC}^0$). In both cases the constructed circuits act on the residual-stream representation of the padded input, which has polynomially many positions of width $\texttt{poly}(N)$ bits.

**Step 3: Looping $C_N^B$ via Lem. 5.1.** The composition $(\mathcal{T}_N^B)^{r(N)}$ corresponds to iterating $C_N^B$ for $r(N) = \Theta(\log^d N)$ steps. Invoking Lem. 5.1 with depth function $d_B(N) = \Theta(1)$, loop function $r(N) = \Theta(\log^d N)$, and the appropriate gate set produces an FO-uniform circuit family of polynomial size and depth $\mathcal{O}(\log^d N)$ computing $(\mathcal{T}_N^B)^{r(N)}$—an FO-uniform $\texttt{AC}^d$ family in the constant-precision case and an FO-uniform $\texttt{TC}^d$ family in the growing-precision case. The uniformity *tightens* from L-uniform to FO-uniform via the uniformity-collapse (cf. Lem. 5.1), since $\texttt{L} \subseteq \text{FO-uniform } \texttt{AC}^d$ for $d \geqslant 1$.

**Step 4: Composing the three pieces.** The looped circuit sits between $C_N^A$ and $C_N^C$. The result is a serial composition of an FO-uniform $\texttt{AC}^d$ (resp. FO-uniform $\texttt{TC}^d$) family with two L-uniform $\texttt{AC}^0 \subseteq $ FO-uniform $\texttt{AC}^d$ (resp. L-uniform $\texttt{TC}^0 \subseteq$ FO-uniform $\texttt{TC}^d$) families, and these classes are closed under fixed serial composition for $d \geqslant 1$ (Merrill & Sabharwal,

2025a, Lem. 8), so the composed family lies in FO-uniform $\text{AC}^d$ (resp. FO-uniform $\text{TC}^d$). It recognizes the same language $\mathcal{L}$ as $\mathcal{T}_N$, yielding $\mathcal{L} \in$ FO-uniform $\text{AC}^d$ in the constant-precision case and $\mathcal{L} \in$ FO-uniform $\text{TC}^d$ in the growing-precision case.

The argument is agnostic to attention type: Lem. 4.2 is stated and proved for both AHATs and SMATs, and the remaining steps only manipulate the resulting circuit families. Hence the lemma holds for both attention patterns. ∎

**Lemma 5.3** (Lem. 3). *Let $\mathcal{C}, \mathcal{R}$ be language classes and let $\mathcal{L}$ be $\mathcal{C}$-complete under $\mathcal{R}$ reductions. If $\mathcal{L} \in$ L-uniform $\text{LPT}^d_{c,1}$ and L-uniform $\text{LPT}^d_{c,1}$ can compute every $\mathcal{R}$ reduction, then $\mathcal{C} \subseteq$ L-uniform $\text{LPT}^d_{c,1}$. The analogous claim holds for L-uniform $\text{LPT}^d_{1,c}$.*

*Proof.* The proof for L-uniform $\text{LPT}^d_{1,c}$ follows directly from Merrill & Sabharwal, 2025a, Lem. 3 (which gives the construction in the log-precision AHAT regime) combined with Lem. 3.1 of this paper (which lifts it to log-precision SMATs while preserving L-uniformity). We therefore focus on the L-uniform $\text{LPT}^d_{c,1}$ case, where the main adaptation is the absence of the layer hash norm at constant precision; we adapt Merrill & Sabharwal's (2025a) construction step-by-step.

Fix any language $\mathcal{L}' \in \mathcal{C}$. By the assumed completeness of $\mathcal{L}$ for $\mathcal{C}$ under $\mathcal{R}$ reductions, there is an $\mathcal{R}$ reduction $\mathsf{t} : \Sigma^* \to \Sigma^*$ such that $\boldsymbol{w} \in \mathcal{L}'$ if and only if $\mathsf{t}(\boldsymbol{w}) \in \mathcal{L}$, with $|\mathsf{t}(\boldsymbol{w})| \in \text{poly}(|\boldsymbol{w}|)$ (cf. Def. 5.1). Let $\mathcal{T}_{\mathcal{L}} \in$ L-uniform $\text{LPT}^d_{c,1}$ be the family recognizing $\mathcal{L}$, and let $\mathcal{T}_{\mathsf{t}} \in$ L-uniform $\text{LPT}^d_{c,1}$ be the family computing $\mathsf{t}$ (which exists by the second precondition: L-uniform $\text{LPT}^d_{c,1}$ recognizes the language $\mathcal{L}_{\mathsf{t}}$ of Def. 5.2). We construct an L-uniform $\text{LPT}^d_{c,1}$ family $\mathcal{T}$ for $\mathcal{L}'$ by stacking $\mathcal{T}_{\mathcal{L}}$ on top of $\mathcal{T}_{\mathsf{t}}$: The first layers of $\mathcal{T}$ compute $\mathsf{t}(\boldsymbol{w})$ symbol-by-symbol in parallel inside disjoint blocks of padding, and the remaining layers run $\mathcal{T}_{\mathcal{L}}$ on the resulting string.

**Step 1: Layout of the padding space.** Let $N \stackrel{\text{def}}{=} |\boldsymbol{w}|$ and fix a polynomial bound $|\mathsf{t}(\boldsymbol{w})| \leqslant N^c$ on the reduction output (with $c \in \mathbb{N}$ depending on $\mathsf{t}$). Suppose $\mathcal{T}_{\mathsf{t}}$ uses at most $N^K$ padding positions to compute one output symbol $\mathsf{t}(\boldsymbol{w})_n$ on input $(\boldsymbol{w}, \text{B}(n))$; for sufficiently large $N$ we upper-bound this by $N^{K+1}$, and for small $N$ we assume $\mathcal{T}_{\mathsf{t}}$ uses a finite lookup table and no padding. The constructed transformer $\mathcal{T}$ divides its padding space into $N^c$ **blocks** of size $N^{K+1}$ each, so that block $n \in [N^c]$ holds the workspace for computing $\mathsf{t}(\boldsymbol{w})_n$. The total padding is polynomial: $N^{c+K+1} \in \text{poly}(N)$.

**Step 2: Identifying the block index $n$ at every position (simulating the role of the layer hash norm).** Merrill & Sabharwal's (2025a) construction at log-precision uses the *layer hash norm* to compute, at every padding position $n'$, the index $n \stackrel{\text{def}}{=} \lfloor (n' - N)/N^{K+1} \rfloor$ of the block containing $n'$. This identifier is what tells each padding position which output symbol of $\mathsf{t}$ its block is supposed to compute, and it is also what lets attention be restricted to a single block. At constant precision the layer hash norm is unavailable, and the transformer cannot itself compute $\lfloor (n' - N)/N^{K+1} \rfloor$ on the fly. However, this quantity depends only on $n'$ and $N$, both visible to the PE machine $\mathcal{M}_2$ of Def. A.5. We therefore extend the PE at every position $n'$ with the additional logspace-computable fields

$$\boldsymbol{p}_{\text{blk}}(n', N) \stackrel{\text{def}}{=} \text{B}^{\pm}\big(\lfloor (n' - N)/N^{K+1} \rfloor\big), \qquad \boldsymbol{p}_{\text{off}}(n', N) \stackrel{\text{def}}{=} \text{B}^{\pm}\big((n' - N) \mod N^{K+1}\big), \tag{39}$$

encoding the block index and within-block offset; both are computable in $\mathcal{O}(\log N)$ space since division and modulo on $\mathcal{O}(\log N)$-bit numbers are in L. This precomputation injects *at the time of PE computation* exactly the information the layer hash norm would have provided in the log-precision construction on the fly.

**Step 3: Stage 1—computing the reduction in parallel across blocks.** Using the per-position block index from Step 2, every position inside block $n$ can recognize itself as belonging to block $n$ and can read off the block-local offset. Attention can therefore be routed within a single block by matching block indices: By Svete & Sabharwal, 2026, Lem. D.7, a single attention head can be made to attend exclusively to positions whose PE block index equals a query block index, ignoring all other positions. This lets us simulate $\mathcal{T}_{\mathsf{t}}$ inside block $n$: The input symbols $w_1 \cdots w_N$ are addressed unconditionally, the $N^{K+1}$ padding positions of block $n$ play the role of $\mathcal{T}_{\mathsf{t}}$'s padding tape, and the query index $n$ is available from $\boldsymbol{p}_{\text{blk}}$ at every position in block $n$. Running this in parallel inside every block uses the same fixed set of layers (every block executes the same computation on different binary inputs $\text{B}(n)$), so $\mathcal{T}_{\mathsf{t}}$'s contribution to $\mathcal{T}$'s depth is exactly $\mathcal{T}_{\mathsf{t}}$'s own depth. After Stage 1, the final position of block $n$ contains the symbol $\mathsf{t}(\boldsymbol{w})_n$.

**Step 4: Stage 2—recognizing $\mathsf{t}(\boldsymbol{w}) \in \mathcal{L}$ with $\mathcal{T}_{\mathcal{L}}$.** We now run $\mathcal{T}_{\mathcal{L}}$ on top of Stage 1's output. The string $\mathsf{t}(\boldsymbol{w})$ that $\mathcal{T}_{\mathcal{L}}$ consumes is spread across the $N^c$ block-final positions: Position $n' \stackrel{\text{def}}{=} N + n \cdot N^{K+1} + (N^{K+1} - 1)$ holds $\mathsf{t}(\boldsymbol{w})_n$. We modify every attention head of $\mathcal{T}_{\mathcal{L}}$ in exactly the way Merrill & Sabharwal's (2025a) construction does: Add a large negative

bias to the attention score of any key position that is *not* block-final, so that each head attends only over the $N^c$ block-final positions and effectively sees the string $\mathtt{t}(\boldsymbol{w})$. Whether a position is block-final is a function of $n'$ and $N$ alone, so it can also be precomputed as a binary PE flag $\boldsymbol{p}_{\mathrm{end}}(n', N) \in \{0, 1\}$ by $\mathcal{M}_2$ in logspace. Likewise, $\mathcal{T}_{\mathcal{L}}$'s PEs are reinterpreted: At block-final position $n'$ we expose $\boldsymbol{p}_{\mathcal{T}_{\mathcal{L}}}(n, N^c)$ (with $n$ read from $\boldsymbol{p}_{\mathrm{blk}}$), so that $\mathcal{T}_{\mathcal{L}}$ sees the same PE pattern it would on a length-$N^c$ input. Since $\mathcal{T}_{\mathcal{L}} \in$ L-uniform $\mathsf{LPT}_{\mathrm{c},1}^{\mathrm{d}}$, the modified $\mathcal{T}_{\mathcal{L}}$ contributes exactly $\mathcal{T}_{\mathcal{L}}$'s depth, and the resulting computation accepts if and only if $\mathtt{t}(\boldsymbol{w}) \in \mathcal{L}$, i.e., $\boldsymbol{w} \in \mathcal{L}'$.

**Step 5: L-uniformity and resource budget.** By construction, $\mathcal{T}$'s parameter matrices are obtained by gluing the matrices of $\mathcal{T}_{\mathrm{t}}$ and $\mathcal{T}_{\mathcal{L}}$ (both L-uniform) with constant-size routing weights for the block-matching and block-final attention biases, so $\mathcal{M}_1$ remains logspace. The PE machine $\mathcal{M}_2$ emits, in addition to $\mathcal{T}_{\mathrm{t}}$'s and $\mathcal{T}_{\mathcal{L}}$'s PEs, the signed-binary block index $\boldsymbol{p}_{\mathrm{blk}}$, the within-block offset $\boldsymbol{p}_{\mathrm{off}}$, and the block-final flag $\boldsymbol{p}_{\mathrm{end}}$; all three are computable in $\mathcal{O}(\log N)$ space. The depth, width, precision, and loop count of $\mathcal{T}$ are the sum of those of $\mathcal{T}_{\mathrm{t}}$ and $\mathcal{T}_{\mathcal{L}}$ plus a constant overhead for the routing layers; the two looped blocks of $\mathcal{T}_{\mathrm{t}}$ and $\mathcal{T}_{\mathcal{L}}$ fit into the multi-block formulation of Def. A.6, each looped $\Theta(\log^d N)$ times, or equivalently merge into a single block run for $r_{\mathrm{t}}(N) + r_{\mathcal{L}}(N) = \Theta(\log^d N)$ iterations under a one-bit phase flag selecting which sub-block executes. Either way, $\mathcal{T} \in$ L-uniform $\mathsf{LPT}_{\mathrm{c},1}^{\mathrm{d}}$. Finally, the Stage 1 construction builds an SMAT that, by the focusing argument of Lem. B.5, behaves identically to an AHAT; the same applies to Stage 2 via $\mathcal{T}_{\mathcal{L}}$. The lemma therefore holds for both attention patterns, completing the constant-precision case. ∎

**Theorem C.1** (Analog of Merrill & Sabharwal, 2025b, Thm. 2). *There exists an L-uniform $\mathsf{LPT}_{\mathrm{c},1}^1$ transformer family $\{\mathcal{T}_N\}_{N \in \mathbb{N}}$ such that $\mathcal{T}_N$ solves connectivity on (directed or undirected) graphs over $N$ vertices: Given the $N \times N$ adjacency matrix of a graph $\mathcal{G}$, $N^3$ padding positions, and $s, t \in [N]$ in binary, $\mathcal{T}_N$ checks whether $\mathcal{G}$ has a path from vertex $s$ to vertex $t$.*

*Proof.* We consider a directed graph $\mathcal{G}$ over $N$ vertices and follow the construction from Merrill & Sabharwal, 2025b, Thm. 2. We again only highlight the differences in the construction.

Let $\boldsymbol{A} \in \{0, 1\}^{N \times N}$ be $\mathcal{G}$'s adjacency matrix. The input to the transformer is a string over the alphabet $\Sigma \overset{\text{def}}{=} \{0, 1, \square, \&\}$, where $\square$ is the padding symbol and $\&$ is a dedicated *separator* symbol used to delimit the binary encodings of the source and target nodes; the adjacency matrix $\boldsymbol{A}$ occupies $N^2$ positions over $\{0, 1\}$, followed by $N^3$ padding positions $\square$, and finally the source and target nodes $s, t \in \{1, \ldots, N\}$. In contrast to Merrill & Sabharwal (2025b, Thm. 2), $s$ and $t$ are represented in binary with $\lceil \log N \rceil$ bits each:

$$\underbrace{A_{1,1} \ldots A_{N,N}}_{N^2} \ \underbrace{\square \ldots \square}_{N^3} \ \& \ \mathtt{B}(s) \ \& \ \mathtt{B}(t) \tag{40}$$

In contrast to Merrill & Sabharwal (2025b, Thm. 2), we cannot rely on the layer hash norm to identify positions. To account for that, we provide the transformer with the following PEs:

$$\mathtt{PE}(n, N) \overset{\text{def}}{=} \begin{pmatrix} \mathtt{B}^{\pm}(n) \\ \mathtt{B}^{\pm}(n \mod N) \\ \mathtt{B}^{\pm}(\lfloor n/N \rfloor) \\ \mathtt{B}^{\pm}(n') \\ \mathtt{B}^{\pm}(n' \mod N) \\ \mathtt{B}^{\pm}(\lfloor n'/N^2 \rfloor) \end{pmatrix} \in \{0, 1\}^{\mathcal{O}(\log(N))}. \tag{41}$$

where $n' \overset{\text{def}}{=} \max(0, n - N^2)$. $\mathtt{B}(n)$ denotes the binary encoding of $n$ using $\lceil \log N \rceil$ bits and $\mathtt{B}^{\pm}(n)$ denotes the signed binary encoding $2\mathtt{B}(n) - \boldsymbol{1}$, where $\boldsymbol{1}$ is the $\lceil \log N \rceil$-dimensional vector of all ones. These PEs can be computed in logspace.

**Initial layers: Reading the source and target.** The transformer $\mathcal{T}_N$ first uses $\lceil \log N \rceil + 1$ layers to read the binary encodings of the source and target nodes $s$ and $t$ stored in the string into a single dimension of the residual stream, using the L-uniform construction from Lem. B.7. After these layers, the residual stream at the dedicated positions for $s$ and $t$ contains $\mathtt{B}(s)$ and $\mathtt{B}(t)$, respectively, in a single coordinate block; combined with the PEs, every position can now refer to both endpoints of the requested path.

**Repeated layers: Iteratively doubling reachability.** The transformer maintains, in its residual stream, two families of predicates indexed by $\ell \in \{0, 1, \ldots, \lceil \log N \rceil\}$:

(1) $B_\ell(i, j) \in \{0, 1\}$, stored at the input position with coordinates $(i, j)$ (one of the first $N^2$ positions), encoding whether $\mathcal{G}$ has a path of length at most $2^\ell$ from $i$ to $j$.
(2) $C_\ell(i, k, j) \in \{0, 1\}$, stored at the padding position with coordinates $(i, k, j)$ (one of the $N^3$ padding positions), encoding whether $\mathcal{G}$ has paths of length at most $2^{\ell-1}$ from $i$ to $k$ *and* from $k$ to $j$.

The base predicate $B_0(i, j) = \boldsymbol{A}(i, j) \vee \mathbb{1}\{i = j\}$ is computed at every input position $(i, j)$ in a single layer: Position $(i, j)$ already holds the adjacency bit $\boldsymbol{A}(i, j)$, and the test $i = j$ is a function of the PE coordinates stored at the position. The iterative step then alternates between two layers, each of which applies once per $\ell \in \{1, \ldots, \lceil \log N \rceil\}$:

1. **Computing $C_\ell$ in the padding positions.** A padding position with coordinates $(i, k, j)$ uses two attention heads to retrieve $B_{\ell-1}(i, k)$ from the input position $(i, k)$ and $B_{\ell-1}(k, j)$ from the input position $(k, j)$. Both retrievals are facilitated by binary PEs: Head 1 attends with query $\mathsf{B}^\pm(i \cdot N + k)$ and head 2 with query $\mathsf{B}^\pm(k \cdot N + j)$, both against keys $\mathsf{B}^\pm(i' \cdot N + j')$ at input position $(i', j')$. Lemmata B.3 and B.5 ensure each head focuses on the unique target input position in $\mathbb{F}_b$ arithmetic. The MLP then computes $C_\ell(i, k, j) = B_{\ell-1}(i, k) \wedge B_{\ell-1}(k, j)$ from the retrieved bits.
2. **Computing $B_\ell$ in the input positions.** An input position with coordinates $(i, j)$ accepts as $B_\ell(i, j)$ iff there exists $k$ with $C_\ell(i, k, j) = 1$, i.e., $B_\ell(i, j) = \bigvee_{k=1}^N C_\ell(i, k, j)$. We compute this disjunction by detection (Lem. B.6): One attention head at the input position $(i, j)$ attends over the $N$ padding positions $(i, k, j)$ for $k \in [N]$ (selected via the PE block $\mathsf{B}^\pm(n')$ that exposes the full padding position index) with value $C_\ell(i, k, j)$ encoded as a one-hot vector; the aggregated value is positive iff some $k$ has $C_\ell(i, k, j) = 1$, and the MLP thresholds it to recover $B_\ell(i, j) \in \{0, 1\}$.

Each pair of layers doubles the reachability radius, so after $\lceil \log N \rceil$ iterations every reachable pair is captured by $B_{\lceil \log N \rceil}$. Both the initial $\lceil \log N \rceil + 1$ layers that read $s, t$ via Lem. B.7 and the $\lceil \log N \rceil$ doubling iterations are obtained by looping a constant-depth block (depth 1 for the reader and depth 2 for the doubler), with the iteration counter $\ell$ tracked in a dedicated residual-stream coordinate. Both blocks fit into the multi-block formulation of Def. A.6, each looped $\mathcal{O}(\log N)$ times; equivalently, the two blocks can be merged into a single block run for $\mathcal{O}(\log N)$ iterations by adding a one-bit phase flag to the residual stream and selecting which sub-block to execute via the MLP. Either way, the construction lies in $\mathsf{LPT}^1_{c,1}$.

**Final layer: Reading off $B_{\lceil \log N \rceil}(s, t)$.** Once $B_{\lceil \log N \rceil}$ has been populated at every input position, a single additional layer at the last position reads off the answer: It issues an attention head with query $\mathsf{B}^\pm(s \cdot N + t)$ against the keys $\mathsf{B}^\pm(i \cdot N + j)$ stored at input positions $(i, j)$, attending uniquely to the position with coordinates $(s, t)$, and copies its $B_{\lceil \log N \rceil}$ value into the residual stream of the final symbol. The output layer then accepts if and only if this value is 1, i.e., $\mathcal{G}$ has an $s$-$t$ path.

**Constant precision and attention-type.** Every gadget above uses constant-precision arithmetic. Because the only attention pattern used is a saturated lookup or a binary-detection aggregation, the constructed SMAT behaves identically to an AHAT, so the construction works for both attention types. ∎

**Lemma 5.4** (Lem. 4). $\mathsf{NL} \subseteq$ L-*uniform* $\mathsf{LPT}^1_{c,1}$.

*Proof.* We follow the proof of Merrill & Sabharwal (2025a, Lem. 4). Let $\mathcal{L}$ be the graph connectivity problem, class $\mathcal{C}$ be NL, and class $\mathcal{R}$ be FO. We will show that the preconditions of Lem. 5.3 are met, which will give us $\mathsf{NL} \subseteq$ L-uniform $\mathsf{LPT}^1_{c,1}$:

1. First, graph connectivity is known to be NL-complete under FO reductions (Immerman, 1999).
2. Second, Thm. C.1 shows that log-depth transformers with cubic padding can recognize the graph connectivity problem $\mathcal{L}$, i.e., $\mathcal{L} \in$ L-uniform $\mathsf{LPT}^1_{c,1}$.
3. Finally, Thm. 4.1 gives L-uniform $\mathsf{LPT}^0_{c,1} =$ L-uniform $\mathsf{AC}^0$. Since L-uniform $\mathsf{AC}^0 \supseteq$ FO-uniform $\mathsf{AC}^0 =$ FO (Mix Barrington et al., 1990), L-uniform $\mathsf{LPT}^0_{c,1}$ can recognize languages in FO and can therefore compute FO reductions.

∎

### C.3.1. CIRCUIT EVALUATION

We now introduce the circuit evaluation problem and show its completeness under L reductions. This section adapts Merrill & Sabharwal, 2025a, App. D to our setting.

**Definition C.1** (AC *circuit encoding*)**.** *Let $C$ be an* AC *circuit over $N$ inputs. We define the encoding*

$$\langle C \rangle \overset{\text{def}}{=} \underbrace{\text{X } \ldots \text{ X}}_{N \text{ times}} \circ \langle G_1 \rangle \circ \ \ldots \ \circ \langle G_M \rangle \tag{42}$$

*where* X *is a placeholder symbol for holding the input of the circuit[14] and $\langle G_m \rangle$ for $m \in [M]$ encodes the $m^{th}$ gate with $K$ inputs as:*

$$\langle G_m \rangle \overset{\text{def}}{=} \text{T}(G_m) \ \& \ \text{B}(g_1) \ \# \ \text{B}(m) \ \ldots \ \& \ \text{B}(g_K) \ \# \ \text{B}(m), \tag{43}$$

*where* $\text{T}(G_m) \in \{\text{AND}, \text{OR}, \text{NOT}\}$ *denotes $G_m$'s type,* $\text{B}(g_i)$ *is the binary encoding of the position of the $i^{th}$ argument of the gate $G_m$, and* $\text{B}(m)$ *is the binary encoding of the gate's index $m$.*

**Example C.1.** *The encoding of the circuit $C(x_1, x_2, x_3) = (x_1 \wedge x_2) \vee x_3$ is*

$$\langle C \rangle \overset{\text{def}}{=} \underbrace{\text{X X X}}_{input} \ \text{AND} \ \underbrace{\text{\&000 \#010 \&001 \#010}}_{arguments\ to\ \text{AND}} \ \text{OR} \ \underbrace{\text{\&011 \#100 \&010 \#100}}_{arguments\ to\ \text{OR}}. \tag{44}$$

There are two main differences between Merrill & Sabharwal's (2025a) definitions and ours:

(1) We encode the argument pointers of each gate using their *binary encodings* instead of the unary ones used by Merrill & Sabharwal (2025a). We require this to ensure that a shallow (in our case, log-depth) constant-precision transformer can convert the circuit encoding into the contents of the residual stream. It is not clear how to do this with unary encodings.

(2) Second, we replicate the positions of the gates after the pointer to each argument (prefixed by the special # symbol). This is needed to avoid the use of the layer hash norm to compute the pointer to the argument's gate, which is done by Merrill & Sabharwal (2025a).

**Definition C.2** ($\mathcal{F}$ *circuit evaluation; Def. 15*)**.** *The $\mathcal{F}$ circuit evaluation problem takes as input $(\boldsymbol{w}, \langle C \rangle)$ where $\boldsymbol{w} \in \{0, 1\}^*$ and $\langle C \rangle$ is a serialized circuit, and outputs $C(\boldsymbol{w})$.*

We refer to the special case where $\mathcal{F} = \text{AC}^{\text{d}}$ as the $\text{AC}^{\text{d}}$ **circuit evaluation problem**. We further define **wide-$\text{AC}^{\text{d}}$** $\subseteq \text{AC}^{\text{d}}$ as the class of circuit families $\{C_N\}_{N=0}^{\infty}$ such that there exists some $c$ such that, for large $N$, the depth of $C_N$ is at most $c \log^d N$ and, crucially, the size is *at least* $N^c$. That is, the size (and hence the width) of the circuit is large relative to its depth. We define the corresponding **wide-$\text{AC}^{\text{d}}$ circuit evaluation problem** as a variant of the circuit evaluation problem with $\mathcal{F} = \text{wide-AC}^{\text{d}}$. The $\text{TC}^{\text{d}}$ analogs of these concepts are defined in Merrill & Sabharwal, 2025a, App. D.

The following lemmata show that both $\text{AC}^{\text{d}}$ and wide-$\text{AC}^{\text{d}}$ circuit evaluation are hard for FO-uniform $\text{AC}^{\text{d}}$ under NL reductions. Their proofs are identical to the ones in Merrill & Sabharwal (2025a), except that we replace $\text{TC}^{\text{d}}$ with $\text{AC}^{\text{d}}$ (the proofs are not affected by the differences in circuit serializations, since our serialization is also logspace-computable).

**Lemma C.1** (*Lem. 11*)**.** *For $d \geqslant 1$, $\text{AC}^{\text{d}}$ circuit evaluation is hard for* L*-uniform $\text{AC}^{\text{d}}$ under* L *reductions.*

**Lemma C.2** (*Lem. 12*)**.** *For $d \geqslant 1$, wide-$\text{AC}^{\text{d}}$ circuit evaluation is hard for* L*-uniform $\text{AC}^{\text{d}}$ under* L *reductions.*

**Lemma C.3** (*Constant-precision path; Lem. 13*)**.** *Let $\langle C \rangle$ be the serialization of a depth-$L$ circuit with* AND, OR, *and* NOT *gates over $\boldsymbol{w} \in \{0, 1\}^*$. There exist two* L*-uniform constant-precision log-width transformer families $\{\mathcal{T}_N^{\text{read}}\}_{N \in \mathbb{N}}$ and $\{\mathcal{T}_N^{\text{step}}\}_{N \in \mathbb{N}}$ such that:*

(1) $\{\mathcal{T}_N^{\text{read}}\}_{N \in \mathbb{N}}$ *is in* L*-uniform $\text{LPT}_{\text{c},1}^1$ and converts the input $\boldsymbol{w} \circ \langle C \rangle$ into internal pointer representations in $\mathcal{O}(\log N)$ loop iterations;*

(2) $\{\mathcal{T}_N^{\text{step}}\}_{N \in \mathbb{N}}$ *is a constant-depth* L*-uniform family in $\text{LPT}_{\text{c},1}^0$ that, when unrolled $L$ times on the output of $\mathcal{T}_N^{\text{read}}$, computes $C(\boldsymbol{w})$.*

*Their composition $\mathcal{T}_N^{\text{step}} \circ \mathcal{T}_N^{\text{read}}$, run for $\mathcal{O}(\log N) + L$ total loop iterations, computes $C(\boldsymbol{w})$. In particular, when $L = \mathcal{O}(\log^d N)$ for $d \geqslant 1$, the composed family is in* L*-uniform $\text{LPT}_{\text{c},1}^{\text{d}}$.*

---

[14]Note that, although the first $N$ positions contain placeholders for input strings, the circuit encoding is independent of the input string—since all compatible strings are of the same length, the circuit encoding does not change depending on the input string.

*Proof sketch.* The high-level idea of the construction is similar to that of Merrill & Sabharwal, 2025a, Lem. 7, but we need to adapt it to the constant-precision setting. Let $C$ be a circuit of depth $L$ over $N$ inputs. The simulation of $C$ is split across the two transformers $\mathcal{T}_N^{\text{read}}$ and $\mathcal{T}_N^{\text{step}}$, each responsible for one stage:

1. Stage 1, implemented by $\mathcal{T}_N^{\text{read}}$: Converting the input $\boldsymbol{w} \circ \langle C \rangle$ into internal representations that will allow the transformer to process it. This stage requires $\mathcal{O}(\log N)$ layers.
2. Stage 2, implemented by $\mathcal{T}_N^{\text{step}}$ unrolled $L$ times: Iteratively applying the circuit operations to the internal representations to compute the final output. This stage requires $L$ layers.

Stage 1 converts the binary encodings of the argument pointers stored in the input string $\langle C \rangle$ into binary encodings stored in the residual stream as per the L-uniform construction in Lem. B.7. These pointers allow the model to retrieve the values stored in those positions (once they become available). Importantly, this conversion only has to happen once for all gates and inputs in parallel, even if the inputs have not been computed yet—this is possible since the encoding of the entire circuit is available at the beginning. This means that the computation of pointers adds a fixed overhead of $\mathcal{O}(\log N)$ layers to the simulation.

At the same time, the positions containing the *gate* addresses $\mathrm{B}(m)$ compute their binary encodings in the same way as the input arguments (cf. Lem. B.7), storing the encoding in another designated part of the residual stream. The final position of the input argument can then attend $\lceil \log N \rceil + 1$ positions forward to retrieve the position of the gate it belongs to—this can be done by storing the (signed) binary encodings of both $n$ and $n + \lceil \log N \rceil + 1$ in the PEs. With this, each input argument $g_N$ contains both the pointer to its value as well as the pointer to the gate it belongs to—this will be used at a later stage of the simulation, when each gate has to read its input argument values before computing its own value.

The rest of the proof (Stage 2) closely follows that of Merrill & Sabharwal, 2025a, Lem. 7. In contrast to Merrill & Sabharwal's (2025a) fully uniform transformer, ours is L-uniform; the only aspect of the transformer family that depends on $N$ is the size of the matrices, which needs to grow with the growing PEs. The counters required to construct such matrices can be implemented in log-space, which is why the transformer family is L-uniform. The transformer uses $L$ layers to simulate the circuit layer by layer, using the pointers constructed in Stage 1 to retrieve the values of the input arguments of each gate. The only difference is that, to avoid issues with constant precision, the model computes the AND gates by detecting $0$ among the inputs and the OR gates by detecting a $1$ among the inputs as per Lem. B.6—this is enough to determine the truth value of the gate.

As above, this constructs an SMAT that behaves like an AHAT, meaning that it holds for both types of models. ∎

**Theorem 5.1** (Thm. 2). *For any $d \geqslant 1$:*

$$\text{L-}uniform\ \mathtt{LPT}_{\mathsf{c},1}^{\mathsf{d}} = \text{FO-}uniform\ \mathtt{AC}^{\mathsf{d}} \tag{8a}$$

$$\text{L-}uniform\ \mathtt{LPT}_{1,\mathsf{c}}^{\mathsf{d}} = \text{FO-}uniform\ \mathtt{TC}^{\mathsf{d}}. \tag{8b}$$

*Proof.* Let $\mathcal{L}$ be the wide-$\mathtt{AC}^{\mathsf{d}}$ circuit evaluation problem, $\mathcal{C}$ be the class FO-uniform $\mathtt{AC}^{\mathsf{d}}$, and $\mathcal{R}$ be the class L. We prove the two equalities in turn.

**Part (a): L-uniform $\mathtt{LPT}_{\mathsf{c},1}^{\mathsf{d}}$ = FO-uniform $\mathtt{AC}^{\mathsf{d}}$.** The upper bound L-uniform $\mathtt{LPT}_{\mathsf{c},1}^{\mathsf{d}} \subseteq$ FO-uniform $\mathtt{AC}^{\mathsf{d}}$ follows directly from Lem. 5.2. For the lower bound, the preconditions of Lem. 5.3 are met:

(1) Cor. 5.1 shows that $\mathcal{L} \in$ L-uniform $\mathtt{LPT}_{\mathsf{c},1}^{\mathsf{d}}$ for $d \geqslant 1$. Together with Lem. 5.2, this implies $\mathcal{L} \in$ FO-uniform $\mathtt{AC}^{\mathsf{d}}$.
(2) Lem. C.2 shows that $\mathcal{L}$ is hard for $\mathtt{AC}^{\mathsf{d}}$ under L reductions. Together with step (1), we get that $\mathcal{L}$ is complete for FO-uniform $\mathtt{AC}^{\mathsf{d}}$.
(3) Lem. 5.4 gives us L $\subseteq$ L-uniform $\mathtt{LPT}_{\mathsf{c},1}^{\mathsf{d}}$, meaning that L-uniform $\mathtt{LPT}_{\mathsf{c},1}^{\mathsf{d}}$ can compute any L reduction.

Thus, Lem. 5.3 applies, yielding FO-uniform $\mathtt{AC}^{\mathsf{d}} \subseteq$ L-uniform $\mathtt{LPT}_{\mathsf{c},1}^{\mathsf{d}}$.

Note that, unlike Merrill & Sabharwal (2025a, Thm. 3), which constructs a transformer with depth $\mathcal{O}(\log^d N)$ to simulate $\mathtt{TC}^{\mathsf{d}}$ circuits, our application of Lem. C.3 yields a transformer with depth $\mathcal{O}(\log N + \log^d N)$ to simulate an $\mathtt{AC}^{\mathsf{d}}$ circuit. For $d \geqslant 1$, this reduces to $\mathcal{O}(\log^d N)$.

**Part (b): L-uniform $\mathtt{LPT}_{1,\mathsf{c}}^{\mathsf{d}}$ = FO-uniform $\mathtt{TC}^{\mathsf{d}}$.**

*Upper bound.* For SMATs, L-uniform $\mathtt{LPT}_{1,\mathsf{c}}^{\mathsf{d}} \subseteq$ FO-uniform $\mathtt{TC}^{\mathsf{d}}$ (cf. Lem. 5.2). For AHATs, the same inclusion applies because Lem. 4.2 and Lem. 5.2 both hold for AHATs as well.

*Lower bound.* The matching lower bound FO-uniform $TC^d \subseteq$ L-uniform $LPT_{1,c}^d$ traces the steps outlined in §5. The starting point is Merrill & Sabharwal, 2025a, Thm. 3, which establishes that fully-uniform log-precision AHATs with $\Theta(\log^d N)$ looping simulate FO-uniform $TC^d$:

$$\text{FO-uniform } TC^d \subseteq \text{FO-uniform } LPT_{1,c}^d \text{ for AHATs.} \tag{45}$$

Merrill & Sabharwal's (2025a) proof of Eq. (45) bases on the reduction framework restated in Lem. 5.3: They show that fully-uniform log-precision AHATs can compute every L reduction (via the NL lower bound, the AHAT analog of Lem. 5.4) and can recognize the wide-$TC^d$ circuit evaluation problem (the AHAT/$TC^d$ analog of Cor. 5.1), and they invoke their reduction lemma (the AHAT/log-precision analog of Lem. 5.3) to conclude. Since wide-$TC^d$ circuit evaluation is FO-uniform $TC^d$-complete under L reductions (Merrill & Sabharwal, 2025a, Lem. 12, App. D) (the $TC^d$ analog of Lem. C.2), this yields Eq. (45).

We lift Eq. (45) to L-uniform SMATs in two steps. First, since fully uniform families are a special case of L-uniform families, Eq. (45) gives

$$\text{FO-uniform } TC^d \subseteq \text{L-uniform } LPT_{1,c}^d \text{ for AHATs.} \tag{46}$$

Second, we apply Lem. 3.1 to each AHAT family produced by Eq. (46): The lemma constructs an L-uniform logarithmic-precision SMAT family whose outputs match the AHAT's on every input, uses the same number of layers, and preserves the loop count of the original looped block (the SMAT replacement is layer-by-layer, with the looped block of the AHAT corresponding to the looped block of the SMAT). The width and PE width are preserved up to a constant factor. Hence the resulting SMAT family lies in L-uniform $LPT_{1,c}^d$, transferring the lower bound to SMATs:

$$\text{FO-uniform } TC^d \subseteq \text{L-uniform } LPT_{1,c}^d \text{ for SMATs.} \tag{47}$$

Combining Eqs. (46) and (47) with the upper bound established above yields L-uniform $LPT_{1,c}^d = $ FO-uniform $TC^d$ for both attention types. ∎

