# OpenReview forum: "Revisiting Padded Transformer Expressivity: Which Architectural Choices Matter and Which Don't"
_ICML.cc/2026/Conference — ICML 2026 regular_

### Official Review · Reviewer_Eugm · 2026-03-09

**Soundness:** 2
**Presentation:** 2
**Significance:** 2
**Originality:** 2
**Overall Recommendation:** 4
**Confidence:** 3

**Summary:**

This paper systematically studies the expressivity of polynomially-padded transformers under various architectural choices, including attention type (softmax vs. average hard attention), model width, numeric precision, uniformity, and depth (via looping). The main message is that padded transformers are surprisingly robust to many of these choices—what really matters is (1) whether precision is constant or growing (which determines an AC vs. TC divide) and (2) model depth (via looping). Concretely, the paper shows that L-uniform constant-precision padded transformers equal L-uniform $AC^0$, growing-precision ones equal L-uniform $TC^0$, and looping scales this to $AC^d / TC^d$ in a natural way. These results unify and extend prior work by Merrill & Sabharwal (NeurIPS 2025) on fully-uniform AHATs and London & Kanade (NeurIPS 2025) on L-uniform SMATs.

**Compliance With Llm Reviewing Policy:**

Affirmed.

**Final Justification:**

My concern is mostly solved. Raise to score 4.

**Key Questions For Authors:**

- Can you give an example of a proof technique or construction in this paper that is genuinely new, rather than an adaptation/extension of Merrill & Sabharwal or London & Kanade? The constant-width PE construction (Lemma B.3) seems like one candidate—can you elaborate on its novelty?
- For the looped transformer results, the uniformity weakens from L-uniform to FO-uniform (Lemma 5.2). Is this an artifact of the proof technique, or is there a fundamental reason looped L-uniform transformers cannot achieve L-uniform $AC^d$?
- You mention that "some degree of non-uniformity seems necessary" for SMATs to simulate AHATs (after Lemma 3.1). Do you have any intuition for whether fully uniform SMATs can simulate fully uniform AHATs, or is there a separation?
- Your resource allocation advice (prioritize precision, then use looping) is based on expressivity alone. Do you have any intuition about whether learnability aligns with expressivity here? For instance, are the constructions in your lower bounds learnable by gradient descent?

**Limitations:**

The paper is purely theoretical and the authors do acknowledge this—they explicitly note they "only consider expressivity" and don't make claims about learnability (Section 6). The restriction to polynomial padding is a significant idealization, and the paper doesn't address whether the theoretical equivalences hold under more realistic padding budgets. The connection to practical transformers is weakened by the use of fixed-point arithmetic, L-uniform families, and the lack of any experiments. Conjecture 6.1 about unpadded transformers is interesting but unresolved. Overall, a relevant question discussed by the study. The authors strive to examine a broad area, but the incremental nature over two recent NeurIPS 2025 papers and absence of empirical grounding limit the contribution.

**Strengths And Weaknesses:**

### Strengths
- S1: Clean and comprehensive unification. The paper fills in essentially all the missing cells in Figure 1, which is a nice contribution. Prior work left open the placement of L-uniform AHATs, constant-width SMATs, and several other intermediate regimes. The paper ties these together through a combination of new lower bounds (e.g., Lemma 4.1 showing constant-width log-precision transformers can simulate L-uniform $TC^0$) and upper bounds (Lemma 4.2), creating a fairly complete picture. The result that precision can compensate for width (Corollary 4.1) is a neat formal statement that hadn't been shown before.
- S2: The looping extension is natural and satisfying. Extending the constant-depth results to the looped setting (Section 5) and showing the expressivity scales analogously to circuits $(AC^d$ and $TC^d)$ is a valuable contribution. The uniformity collapse result—that L-uniform looped transformers are upper-bounded by FO-uniform classes—is a nice structural insight. The three-step reduction-based technique for the constant-precision lower bound (Lemma 5.3 -> Lemma 5.4 -> Corollary 5.1 -> Theorem 5.1) is cleanly presented.
### Weaknesses
- W1: Incremental over the two main prior works. This is my main concern. The paper is heavily built on top of Merrill & Sabharwal (2025a) and London & Kanade (2025), both NeurIPS 2025. Many of the key techniques—the reduction framework (Def. 5.1, 5.2, Lem. 5.3), circuit evaluation approach, the looping-to-depth argument (Lem. 5.1)—are directly adapted from Merrill & Sabharwal. The constant-depth lower and upper bounds extend London & Kanade's constructions. The SMAT-AHAT bridge (Lemma 3.1) relies on Yang et al. (2025b)'s temperature scaling result. The individual proof steps often state things like "the proof follows [prior work] with one difference" (see proofs of Lemma 4.1, 4.2, 5.1, 5.2, 5.3 in the appendix). While I appreciate that putting these together is nontrivial and the final picture is clean, the novelty of each individual piece feels limited. It reads more like a thorough "closure" of an existing research program than a paper introducing fundamentally new ideas.
- W2: The "robustness" message is somewhat expected. The paper's central selling point is that padded transformers are "surprisingly robust" to architectural choices. But given that padding provides polynomial extra space for computation—which already intuitively allows simulating poly-size circuits—it's not that surprising that details like attention type or width don't matter much once you have enough volume. The volume condition V(N) = Omega(log N) is a basic requirement just to index positions, and beyond that, the circuits do the heavy lifting. The interesting direction (transformers with insufficient volume, Conjecture 6.1) is acknowledged but left open, and the paper even admits that "describing the expressivity of transformers with insufficient volume is difficult" (point (5) in the intro).
- W3: Limited discussion of what's NOT covered. The paper restricts to polynomial padding throughout, but doesn't deeply engage with what happens with sub-polynomial padding (e.g., linear or n log n padding). The discussion briefly mentions physically-realizable circuits (Prada & Mali, 2025) as a connection for limited padding, but this is quite thin. Also, the paper focuses on language recognition (binary classification) and doesn't discuss implications for language modeling or generation, which is what transformers are actually used for in practice. The gap between the formal model and reality is acknowledged but not seriously addressed.

---

> ### Author Rebuttal · Authors · 2026-03-31
>
> Thank you for the thorough and constructive review. We are happy that you see the value of the unification and the utility of the looping results. We address your points below.
>
> **W1.** We emphasize that the synthesis itself is the intellectual contribution. You note that individual proof steps follow prior work, but the challenge and the novelty lie in identifying which prior techniques compose and under what assumptions. This required navigating the interactions between parameters in a way that no prior work attempted. The extensive appendix with proofs---including for results that may seem intuitive in hindsight---reflects this complexity of formalization and connection. We also note that several technical novelties were necessary, including the precision-compensates-for-width result (Cor. 4.1), the error characterization for translating AHAT lower bounds to SMATs (Lem. B.3), and the finite-precision looping construction (Lem. B.6). In this sense, the idea itself that such a concise description of transformer expressivity is possible is a novel, unexpected outcome. Relying on existing results also makes our results recognizable. Relatedly, we do not invent abstractions (adding to their growing number); rather, we unify existing ones that have thus far existed independently.
>
> **W2.** We do not see the robustness as obvious: it is not self-evident that transformers would be able to use the additional space for useful computations and thus capture an entire circuit class. The volume condition alone does not prescribe *how* extra capacity can be used---it is a priori possible that some architectural choices waste the available computation or that certain attention mechanisms cannot exploit the available space.
>
> **W3.** We focus on polynomial padding as this is standard in the literature. A logarithmic or constant number of padding tokens does not increase expressivity in a circuit-class sense ([5], Thm. 4). The connection to linearly-padded models via realizable circuits is promising, but the framework is new and not well characterized. We believe that the diversity of models and the uniformity of results bring the theoretical results closer to practical models: as the idealizations are equivalent, results seem more likely to extend to practical models. Regarding language modeling: our results generalize to circuit *functions* (e.g., $FAC^d, FTC^d$), and we view recognition as a standard sub-task for computing correct symbol probabilities.
>
> **Q1.** We emphasize the following:
> (a) Lem. B.3, yielding Cors. B.1 and 4.1. It characterizes errors from fixed-point arithmetic over unit-length PEs, providing a general bridge from AHAT lower bounds to growing-precision SMATs. Prior SMAT results relied on exact computations with log-length binary PEs, while AHAT results bypass this issue entirely (since the attention weight gaps are irrelevant). Similar techniques can be used for any fixed-point operations in a transformer (possibly different PEs, etc.) to translate AHAT results to SMAT ones.
> (b) Lem. B.6 shows how to read and store a circuit encoding into the residual stream at finite precision, enabling the looping results.
> (c) Sec. C.3.1: the circuit evaluation construction required a modified encoding (via Lem. B.6) and an additional $\log(N)$ looping factor absent from log-precision constructions.
>
> **Q2** It is not an artifact. It follows from the uniformity collapse [6]: $L ⊆ FO\ uniform\ AC^1$, so L-uniform constructions themselves can be simulated in $FO\ uniform\ AC^1$ directly, making the looser uniformity unnecessary.
>
> **Q3.** We conjecture a separation, since temperature scaling [1] appears necessary for simulating hard attention with fixed parameters, and it introduces a minimal amount of non-uniformity. Whether this constitutes "non-uniform" is contingent on the definition.
>
> **Q4.** This is an exciting open question. Transformers often learn constructions different from theoretical ones even on tasks they solve perfectly [2], so we view our results as existence proofs. Encouragingly, empirical work finds that looping improves performance [3,4,*inter alia*], and more expressive architectures scale better empirically [7].
>
> [1] Yang A., Strobl L., Chiang D., and Angluin D. Simulating hard attention using soft attention.
> [2] Wen K., Li Y., Liu B., and Risteski A. Transformers are uninterpretable with myopic methods: A case study with bounded Dyck grammars.
> [3] Geiping J., McLeish S., Jain N., Kirchenbauer J., Singh S., Bartoldson B. R., Kailkhura B., Bhatele A., and Goldstein T. Scaling up test-time compute with latent reasoning: A recurrent depth approach.
> [4] Fan Y., Du Y., Ramchandran K., and Lee K. Looped transformers for length generalization.
> [5] Merrill W. and Sabharwal A. A little depth goes a long way: The expressive power of log-depth transformers.
> [6] Merrill W. and Sabharwal A. Exact expressive power of transformers with padding.
> [7] OLMo Hybrid Team. OLMo Hybrid: From Theory to Practice. 2026.

---

> > ### Author Rebuttal · Reviewer_Eugm · 2026-04-02
> >
> > Thanks. Raised my score to weak accept.

---

### Official Review · Reviewer_7H1P · 2026-03-10

**Soundness:** 3
**Presentation:** 2
**Significance:** 3
**Originality:** 2
**Overall Recommendation:** 4
**Confidence:** 2

**Summary:**

This paper establishes exact equivalences between polynomially-padded transformers and boolean circuit complexity classes ($AC^0$, $TC^0$, $AC^d$, $TC^d$). The authors demonstrate that as long as a transformer has sufficient "volume" ($\Omega(\log N)$), its expressivity is surprisingly robust to architectural choices like attention type and width. The core finding is that expressivity is primarily determined by numeric precision and model depth, mathematically proving that precision can compensate for width

**Compliance With Llm Reviewing Policy:**

Affirmed.

**Key Questions For Authors:**

1.Your gadgets exploit fixed-point non-associativity. Would these equivalences collapse under standard IEEE 754 floating-point arithmetic?2.You construct a specific 2D positional encoding to prove constant-width bounds. Would the $TC^0$ / $AC^0$ equivalences hold if restricted to standard RoPE or sinusoidal encodings?3. Polynomial padding is a strong assumption. What happens to the complexity class if we restrict the model to linear padding (O(N))?

**Limitations:**

yes

**Strengths And Weaknesses:**

Soundness:The theoretical framework is rigorous, effectively utilizing fixed-point arithmetic to model transformer operations.The proofs rely heavily on customized 2D positional encodings and the non-associativity of fixed-point rounding, which deviate from standard empirical setups.
Presentation:Figure 1 provides an excellent taxonomy of transformer variants and their circuit class equivalents.The notation (e.g., $LPT_{b,D}^d$) is dense and slightly overwhelming for non-experts.
Significance: Unifies highly fragmented theoretical literature and provides valuable insights into the scaling behavior of looped transformers.The foundational assumption of polynomial padding ($poly(N)$) is practically prohibitive for real-world autoregressive language models.
Originality:Establishing exact equivalences rather than loose upper bounds is a strong contribution. The formal proof that precision compensates for width is highly novel.

---

> ### Author Rebuttal · Authors · 2026-03-31
>
> Thank you for the positive evaluation of the paper’s contributions and clarity, and for recognizing its impact\! We address your points below.
>
> 1. Using fixed-point arithmetic is standard in the theoretical literature (\[1, 2, 3\], inter alia). As prior fixed-precision constructions, our constructions use rounding not merely as a source of error but as a tool—specifically, to introduce non-linearity. Regarding IEEE 754 floating-point: it is itself non-associative \[6, p. 8\], so the core mechanism behind our constructions survives. Thus, lower bounds hold for floating-point models as well. In contrast, establishing upper bounds for floating-point models remains open and is an important direction for future work, as it would bring the results closer to practical implementations. The only work on this is \[1\]'s brief study of this question. They conclude that drawing the same conclusions as with fixed-point arithmetic would be difficult.
> 2. The specific 2D encodings are used because they allow isolating individual positions by concentrating attention weight, and encoding other (L-computable) information about the input positions directly into the positional representation. Similar PEs have been used for AHATs (\[4\]). Importantly, these encodings can be used \*in addition to\* standard ones like RoPE or sinusoidal PEs—they are not a replacement. Our results thus suggest that such auxiliary positional encodings might be beneficial for certain kinds of reasoning tasks, which would be worth exploring empirically. We are not aware of analogous constructions using RoPE or sinusoidal PEs alone.
> 3. The expressivity of transformers with limited padding is an interesting open question. The motivation for the polynomial number of padding symbols comes from the connection to circuit classes with polynomially large circuits: Polynomial padding provides polynomially many (parallel) computational units, which can be used analogously to gates in (uniform) circuits. This is the core intuition behind the constructions connecting transformers to entire (uniform) circuit classes.
>    If we think of transformers as circuits with “roughly” a quadratic number of gates (due to the quadratic complexity of the attention mechanism), linear padding adds an additional quadratic amount of computation. This increase is, however, difficult to link to increases in expressivity with standard circuit complexity classes, which are defined w.r.t. poly-size circuits. Note, however, that a logarithmic or constant number of padding tokens does not increase expressivity; at least not in a sense “measurable” by a circuit class (see, e.g., \[5\], Thm. 4 and \[1\], Fig. 10).
>
> \[1\] Li, Z., Liu, H., Zhou, D., and Ma, T. Chain of thought empowers transformers to solve inherently serial problems. In ICLR, 2024\. URL [https://openreview.net/forum?id=3EWTEy9MTM](https://openreview.net/forum?id=3EWTEy9MTM).
>
> \[2\] London, C. and Kanade, V. Pause tokens strictly increase the expressivity of constant-depth transformers. In NeurIPS, 2025\. [https://openreview.net/forum?id=eG5oh8l1WZ](https://openreview.net/forum?id=eG5oh8l1WZ).
>
> \[3\] Saunshi, N., Dikkala, N., Li, Z., Kumar, S., and Reddi, S. J. Reasoning with latent thoughts: On the power of looped transformers. In ICLR, 2025\. [https://openreview.net/forum?id=din0lGfZFd](https://openreview.net/forum?id=din0lGfZFd).
>
> \[4\] Merrill, W. and Sabharwal, A. The expressive power of transformers with chain of thought. In ICLR, 2024\. [https://openreview.net/forum?id=NjNGlPh8Wh](https://openreview.net/forum?id=NjNGlPh8Wh).
>
> \[5\] Merrill, W. and Sabharwal, A. A little depth goes a long way: The expressive power of log-depth transformers. In NeurIPS, 2025\. [https://openreview.net/forum?id=5pHfYe10iX](https://openreview.net/forum?id=5pHfYe10iX).
>
> \[6\] Chen, D.H.C. The Removal/Demotion of MinNum and MaxNum Operations from IEEE 754™-2018. https://grouper.ieee.org/groups/msc/ANSI\_IEEE-Std-754-2019/background/minNum\_maxNum\_Removal\_Demotion\_v3.pdf

---

> > ### Author Rebuttal · Reviewer_7H1P · 2026-04-02
> >
> > The authors' rebuttal clarifies my initial concerns, particularly regarding floating-point arithmetic and the 2D positional encodings. While the theoretical analysis is solid, there remain open questions about how idealized assumptions (like polynomial padding) might translate to practical empirical setups. Overall, I keep my positive rating.

---

### Official Review · Reviewer_dDfP · 2026-03-12

**Soundness:** 3
**Presentation:** 2
**Significance:** 2
**Originality:** 2
**Overall Recommendation:** 4
**Confidence:** 3

**Summary:**

This paper continues the transformer expressivity works by introducing both an upper and lower bounds for transformers with paddings, in terms of circuit complexity. The paper proves that padded transformers are robust against attention type, model width and uniformity. An interesting observation is that growing width or precision beyond logarithmic does not increase expressivity.

**Compliance With Llm Reviewing Policy:**

Affirmed.

**Final Justification:**

The rebuttal have addressed most of my concerns.

**Key Questions For Authors:**

Please see Weaknesses.

**Limitations:**

Yes

**Strengths And Weaknesses:**

Strengths:
1. The topic is important and interesting: formal guarantees of transformer expressivity is always helpful for understanding the architecture deeper
2. Both upper and lower bounds are coherent and reasonable, and comparisons with earlier bounds are convincing
3. To the best of my knowledge, padded transformer expressivity is not well-studied yet

Weaknesses:
1. For works like this, precision is always the key assumption and one of the most critical key factors for expressivity. Non-constant precision (e.g. logarithmic with respect to input length) is not possible and therefore not a realistic assumption.
2. Some results in the paper (e.g. Lemma 4.1, 5.2) looks like folklores as the main points are "log-precision transformers are powerful, and they are more powerful than constant-precision transformers"
3. The claim of "growing width or precision beyond logarithmic does not increase expressivity" seems restrained: Lemmas 4.1 and 4.2 only consider transformers with 0 loops, not the entire architecture
4. Writing is confusing at several points. For instance, the definitions of $D(N)$ and $b(N)$ (central to define the volume) were only given in page 2 inside the paragraph; such the central notion should be defined clearly by spelling out $D$ and $b$ rigorously. Below Definition 2.1, "One of our main results reveals that" should point out which main results.

---

> ### Author Rebuttal · Authors · 2026-03-31
>
> Thank you for the thoughtful review! We are happy that you find the topic interesting and that you see its value for understanding transformers. To address your points:
>
> 1. We respectfully disagree that non-constant precision undermines the contribution. First, growing precision (particularly log precision) is a standard assumption in the theoretical literature on transformer expressivity (\[1, 2, 3\]), precisely because it captures core desiderate of the minimal viable uniform transformer: One that can index individual positions (this requires $\\log n$ bits for an input with $n$ tokens) and attend over all of them. The latter, in particular, is a core component of any transformer with a global representation of the string, and we argue that such a pattern is indeed natural, even in practical transformers. Our work's contribution is not to introduce this assumption, but to show that under the same assumption used by multiple prior efforts, precision turns out to be \*the\* dominant factor while other architectural choices (attention type, width) are surprisingly irrelevant. We also note that the draft *also* describes constant-precision transformers.
>
>    Second, the log precision regime is not merely a theoretical convenience—it is practically grounded. Standard practical precision regimes (e.g., float32 in mixed precision for computing attention weights) suffice for accurately describing long context lengths, making log precision viable in practice. For example, for the typical context lengths models are trained for (1M tokens), 32 bits suffices for uniquely representing individual positions. In this sense, our results about log-precision transformers directly speak to real systems.
>
>    Third, fixed-precision results also often underestimate the practical abilities of transformers. For example, transformers with sufficient supervision can represent the parity language, despite it lying outside AC0, which is a known upper bound on fixed-precision transformers  \[1, 4\]. This suggests that our framework, if anything, gives a conservative picture of what real transformers can do.
>
>    Altogether, our results provide a comprehensive overview of how modeling choices affect expressivity and theory-based architectural guidelines—for example, motivating the use of sufficient precision, especially for the computation of attention weights.
> 2. We want to clarify that our main results are not simply that "log precision is more powerful than constant precision." Rather, the central finding is that, under polynomial padding, many architectural choices that theoreticians commonly believe to matter, e.g., attention type, turn out to be irrelevant within each precision regime. This is not folklore; prior to our work, it had not been formalized, and it was not obvious it would hold uniformly across so many architectural variants (e.g., soft vs. average-hard attention). We appreciate that the results may feel intuitive in hindsight, but we see this as the mark of a successful unification rather than evidence against novelty.
>    Moreover, the specific result that log precision can compensate for log width (Cor. 4.1) is, to our knowledge, entirely new: All previous work on the relationship between precision and width combines log precision with log width to describe log-precision transformers.
> 3. We believe there might be a misreading of the scope of our results. Lem. 4.1 and 4.2 characterize non-looped (constant-depth) transformers, showing that growing width or precision beyond log does not increase expressivity in that setting. *The same* conclusion holds for looped transformers as well: The expressivity gains from looping come from the increased depth, not from super-log width or precision. Log precision already suffices to capture the full power of the relevant circuit classes in the looped setting (Lem. 5.2). In other words, the claim that “growing width or precision beyond log does not increase expressivity” holds across our entire framework. The key factor that does increase expressivity beyond the constant-depth case is depth (via looping), not width or polynomial precision.
> 4. Thank you for the suggestions on how to improve clarity\! We will add the definitions in a separate definition environment. We note that both are standard and established in the literature on transformer expressivity (as functions of $N$). We will also point to the concrete result we refer to in the Preliminaries (Lem. 4.2).
>
> \[1\] London, C., Kanade, V. Pause tokens strictly increase the expressivity of constant-depth transformers.
> \[2\] Merrill, W. and Sabharwal, A. A little depth goes a long way: The expressive power of log-depth transformers.
> \[3\] Saunshi, N., Dikkala, N., Li, Z., Kumar, S., and Reddi, S. J. Reasoning with latent thoughts: On the power of looped transformers.
> \[4\] Yang A., Chiang, D., and Angluin, D. Masked hard-attention transformers recognize exactly the star-free languages, 2024\.

---

> > ### Author Rebuttal · Reviewer_dDfP · 2026-04-03
> >
> > My concerns are mostly solved. I have updated the score to weak accept.

---

### Official Review · Reviewer_Ee58 · 2026-03-13

**Soundness:** 2
**Presentation:** 3
**Significance:** 3
**Originality:** 2
**Overall Recommendation:** 5
**Confidence:** 3

**Summary:**

This work studies the expressive power of (polylog-looped, possibly non-causal) polynomially-padded transformers across several combinations of model configurations (namely, depth/looping, width, precision, attention type, and uniformity) established upon circuit complexity theory. The analysis reveals that, given sufficient volume (i.e., width * precision = $\Omega(\log N)$), the main factors separating the expressivity class of the padded transformers are the precision (constant v.s. $\log(N)$) and the depth (constant depth v.s. $\log^d(N)$ looping). Growing width (i.e., hidden dimension) and changing the attention type (either softmax or average-hard) do not change the expressivity class of padded transformers; under polylog looping, weakening uniformity has no benefit as well.

**Compliance With Llm Reviewing Policy:**

Affirmed.

**Final Justification:**

The authors' rebuttal addressed my main concerns. I remain the confidence of my assessment as 3 because I believe more proof details can be filled in from the current manuscript but I did not have any chance to check them all. That said, the proof sketches of Lemmas 5.3 and C.3 seems convincing and showcase the main technical contribution of this work. Hence, I raise my score to 5 as I promised.

**Key Questions For Authors:**

1. Why *polynomially*-padded? I am curious why this work and a previous work on padded transformers (London & Kanade, 2025) are particularly interested in polynomially-padded transformers. Even though it is interesting to read the comparison between polynomially-padded transformers and unpadded transformers in Section 6, I was wondering: “Why not constant- or polylog-paddings?” What makes *polynomially* many padding tokens particularly interesting?
2. Considering the weakness #2(”significance”) I have raised above, I guess Figure 1 tells little information about padded transformers with (poly)logarithmically many layers *without* looping. This might be an interesting future work, yet I believe the paper should mention this, given that there are some existing theoretical works studying the expressivity of log-depth transformers:
    - Sanford et al., Transformers, Parallel Computation, and Logarithmic Depth. ICML 2024.
    - Sanford et al., Understanding Transformer Reasoning Capabilities via Graph Algorithms. NeurIPS 2024.
3. It was remarked under Lemma 3.1 that it is unknown whether *fully-uniform* SMATs can simulate fully-uniform AHATs. Is it still unclear even after introducing $\tt \pm \infty$ (and/or $\tt NaN$) to the $b$-precision number system as practical floating-point numbers? As far as I understand, in Section 2, the set $\mathbb{F}_b$ is certainly the set of finite numbers. (But I guess, although I am not 100% sure, introducing non-finite values to the number system won’t dramatically change the expressivity because the current $\mathbb{F}_b$ is quite expressive enough, e.g., Lemma B.1—am I correct?)

**Limitations:**

Yes

**Strengths And Weaknesses:**

# Strengths

1. Soundness: I cannot find any critical logical errors in the main theorems’ proofs (if I understood the proofs correctly).
2. Presentation: The paper is well organized and contains a clear visual summarization (e.g., Figure 1). It provides a lot of discussions that help readers to thoroughly understand their theoretical contributions.
3. Significance: This work fills in a bunch of the blanks in the expressivity table for (looped) padded transformers, in terms of key model hyperparameters. There are several interesting facts proved in the paper:
    - Corollary 4.1: For L-uniform constant-depth polynomially-padded transformers, growing with logarithmically (remaining precision constant) is never more beneficial than growing precision logarithmically (remaining width constant).
4. Originality: Although many proofs in this work are very similar to a few previous works, all the proofs provide their key technical novelties that enable proving similar results in extended assumptions.

# Weaknesses

1. Presentation: The readability of the paper can be enhanced if it includes some more background about complexity classes, such as Nondet-LOGSPACE($\tt NL$), LOGSPACE($\tt L$), and DLOGTIME($\tt FO$) (and the hierarchy between them: $\tt FO \subset L \subset NL$). Also, the paper itself does not contain a *formal* definition of ‘fully uniform’ transformers. Moreover, mentioning the fact ${\tt TC^d \subset AC^{d+1}}\ (d\ge 1)$ might help readers understand the relevance between different looping counts more comprehensively.
    - A minor typo: Section 5.2, Line 275, Right column, “This lower naturally applies…” $\to$ “This lower bound naturally applies…”
2. Significance: Although the “depth” is mentioned as a key factor to separate expressivity of padded transformers, the word is (mis-)used in a very specific circumstance: it actually means the presence and amount of “looping.” Figure 1 seems to compare constant-depth and polylog-depth padded transformers. It turns out, however, the term “$\Theta (\log^d N)$” in the table represents the number of loopings, as revealed at the end of the introduction.
3. Soundness & Originality: Most of the proofs are done in words, because they are mostly modifications of the proofs made by London & Kanade (2025) and Merrill & Sabharwal (2025a;2025b). Although it is a nice way to shorten the proofs without spending lots of pages, I do not think it is the best way to present the proofs. This is because it may hide some logical connections among technical arguments that complete the proofs, obscuring their verifiability.
    - However, I am willing to increase my score if the authors can provide a full proof of one or two main theorems by explicitly applying the modifications in the current proofs during the rebuttal phase.

---

> ### Author Rebuttal · Authors · 2026-03-30
>
> Thank you for the positive evaluation of the paper’s clarity and contributions\!
>
> ### Weaknesses
> 1. Thank you for the suggestions on how to make the paper more easily understandable. Should the paper be accepted, we will use the additional page to add the suggested definitions and discussion of relevant complexity classes; they would definitely make the paper more self-contained.
> 2. Distinguishing depth and looping is indeed something we were not careful enough about. Thank you for bringing this up. We indeed use “depth” to mean looping, which is a particular, highly uniform way to achieve a desired model depth. We will correct the usage throughout the paper to make this distinction clearer. See also the answer to your question below on how our results apply to “deep transformers”.
> 3. We intentionally kept the proofs short and connected to related work, as it makes the connection to related work clearer by focusing mainly on the changes that have to be made. We felt that spelling out entire proofs would obscure the elegance of the generalizations and complicate understanding the theory. That said, we do see your point of view and will add full proofs of the results that require the largest modifications from the referenced works: Lemma 5.3 on the implementation of reductions with transformers and Lemma C.3 on the simulation of circuits (with the encoding defined in our submission). Hopefully, these full proofs will provide a more complete picture of how our constructions naturally generalize existing ones.
>
> ### Questions
> 1. The motivation comes from the connection to standard circuit classes, which allow polynomially many gates: Polynomial padding provides polynomially many (parallel) computational units, which can be used analogously to gates in (uniform) circuits. This is the core intuition behind the constructions connecting transformers to entire (uniform) circuit classes.
>    This connection to circuit sizes allows us to show that a logarithmic (and thus also a constant) number of padding tokens does not increase expressivity; at least not in a sense “measurable” by a standard circuit class (see, e.g., \[1\], Thm. 4 and \[2\], Fig. 10).
> 2. We did not address the question of transformers with polylogarithmically many independent (not looped) layers. While differences exist, we believe that all our results (lower bounds certainly) generalize to such transformers as long as the layer constructions are sufficiently uniform. For example, for upper bounds, one most likely has to assume that the growing number of independent layers of the transformer can be constructed in an FO-uniform (rather than L-uniform) fashion for all looping results to translate; since looping does not reuse the layers, uniformity collapse would not happen, and the stricter FO-uniformity is probably required. We will add this to the discussion section of the paper. We, however, chose to focus on looped transformers due to their attractive uniformity properties and practical considerations (e.g., parameter-saving nature).
> 3. The inclusion of $\\pm \\infty$ in transformer arithmetic is an interesting consideration and has seen some concurrent work. The exact arithmetic we use does not offer an immediate insight into how infinites would change the results, since, as you observe, we do not use the “sink” values of $\\pm \\infty$. We conjecture that the lower bounds naturally still hold with the extended arithmetic. Upper bounds are, of course, trickier. They are not obvious, but, as you mention, much of the $\\pm \\infty$ behaviour can also be simulated without the values; for example, $\\exp(x) \= B$ and $\\exp(-x) \= 0$ for many sufficiently large $x$, which is one of the core “uses” of the $\\pm \\infty$ values. Moreover, some existing constructions suggest that the “sinking” behavior can, to some degree, be simulated without $\\pm \\infty$ values (\[3\], Lem. D.7). Nevertheless, the relationship would have to be investigated more precisely to determine exact upper bounds.
>
> \[1\] Merrill, W. and Sabharwal, A. A little depth goes a long way: The expressive power of log-depth transformers. In NeurIPS, 2025\. https://openreview.net/forum?id=5pHfYe10iX.
>
> \[2\] Li, Z., Liu, H., Zhou, D., and Ma, T. Chain of thought empowers transformers to solve inherently serial problems. In ICLR, 2024\. URL https://openreview.net/forum?id=3EWTEy9MTM.
>
> \[3\] Svete, A. and Sabharwal, A. On the reasoning abilities of masked diffusion language models. In ICLR, 2026\. [https://openreview.net/forum?id=BVnIsh4Nz1](https://openreview.net/forum?id=BVnIsh4Nz1).

---

> > ### Author Rebuttal · Reviewer_Ee58 · 2026-03-31
> >
> > I thank the authors for their detailed rebuttal.
> >
> > First of all, I am mostly satisfied with the responses. However, as I mentioned in a small bullet point under weakness #3, I promised (and only promised) to raise my score if I could read the full, self-contained proofs of the main results. I believe this is a crucial step to genuinely validate the paper's overall soundness as a reviewer. That said, I also understand that this short rebuttal is insufficient to contain the full details of the proofs. Considering these, as well as my mid-range confidence in my understanding and assessments, I decide to retain my score for now.

---

> > > ### Author Response · Authors · 2026-04-04
> > >
> > > Thank you for reading our response and engaging in a discussion. Here, we would like to expand upon our original response and complete the two proofs mentioned above, as they constitute the core parts of the results concerning looped transformers and are the results that required the largest adaptations from their existing counterparts. We hope this helps provide the picture of how our results translate from existing work. Due to the character limit, the proofs are necessarily concise, but contain all steps from their origin publications.
> > >
> > > ## Proof of Lem. 5.3
> > > Consider $L' \in \mathcal{L}$.
> > > By completeness of $L$, there exists reduction $r$ that maps inputs of $L'$ into inputs of $L$ such that $\boldsymbol{w} \in L'$ if and only if $r(\boldsymbol{w}) \in L$.
> > > We also have that there exists $T_1 \in LPT_{c, l}^d$ that computes the reduction function $r \in \mathcal{R}$ and $T_2 \in LPT_{c, l}^d$ that recognizes $L$.
> > >
> > > The idea is to ``stack'' a modified $T_2$ on top of a modified $T_1$ to obtain $\mathcal{T}$ recognizing $L'$.
> > > Intuitively, given an input $\boldsymbol{w}$, the first set of layers of $\mathcal{T}$ computes $r(\boldsymbol{w})$ token-wise, by computing $r(\boldsymbol{w}, n) = r(\boldsymbol{w})_n$ for every $n$ in parallel.
> > > To this end, we make $|r(\boldsymbol{w})|$ copies of the padding positions needed by $T_1$ and perform the computation of $T_1$ independently for each $r(\boldsymbol{w})_n$.
> > > The second set of layers of $\mathcal{T}$ then checks whether the string $r(\boldsymbol{w})$ produced by the first set of layers is in $L$, which holds if and only if $\boldsymbol{w} \in L'$.
> > > We next make this idea more concrete.
> > >
> > > ##### Compute reduction.
> > > Suppose $T_1 \in LPT_{c, l}^d$ and let $N = |\boldsymbol{w}|$.
> > > Then, on input $(\boldsymbol{w},\texttt{B}(n))$, $T_1$ uses $O(N^K)$ padding positions to compute $r(\boldsymbol{w})_n$, which we upper bound by $N^{K+1}$ for sufficiently large $N$ (for small $N$, we assume $T_1$ instead uses a finite lookup table and no padding).
> > > Since $|r(\boldsymbol{w})|$ is bounded by some polynomial $N^c$, we know that any $n \leq |r(\boldsymbol{w})|$ takes at most $c \log N$ bits to specify, which we upper bound by $c' = N$.
> > >
> > > We then use $I = N^c$ blocks of padding positions, each of size $N^{K+1}$.
> > > A modified $T_1$ uses the $N^{K+1}$ padding positions in block $i$ to consume $(\boldsymbol{w},B(n))$ and compute $r(\boldsymbol{w},i)$.
> > > Specifically, additionally providing $T_1$ with the L-computable PEs $\texttt{B}^{\pm}(\frac{i}{N^{K + 1}})$, $T_1$ can attend to the input symbols and the current block ([1], Lem. D.7).
> > > Thus, block $i$ simulates the computation of $T_1$ on $(\boldsymbol{w},B(n))$ with $N^K$ padding, and the final symbol of block $i$ (at position $i \cdot N^{K+1}$) contains $r(\boldsymbol{w}, i) = r(\boldsymbol{w})_i$.
> > >
> > > ##### Solve complete problem.
> > > We then apply a modified version of $T_2$ that attends only to the positions that hold $r(\boldsymbol{w}, i) = r(\boldsymbol{w})_i$.
> > > By providing $T_2$ with the L-computable PEs $\texttt{B}^{\pm}(i \cdot N^{K+1})$ for $i \in [I]$, $T_2$ can isolate those positions ([1], Lem. D.7) and compute whether $r(\boldsymbol{w}) \in L'$, which is equivalent to recognizing whether $\boldsymbol{w} \in L$.
> > >
> > > ## Proof of Lem. C.3
> > >
> > > We continue from L.1191 (Stage 2).
> > >
> > > The next layer uses the constructed pointers to retrieve the node values stored in those positions.
> > > If the value has already been computed, it is copied; otherwise, the copied value remains empty.
> > >
> > > The main circuit simulation step follows.
> > > Any input gate can now attend to the positions of its input arguments (positions that contain its PE) and retrieve their values from the residual stream.
> > > A single transformer layer can compute the value of the gate by combining the retrieved input values according to the gate's function.
> > > The model computes the AND gates by detecting 0 among the inputs and the OR gates by detecting a 1 among the inputs (Lem. B.5).
> > > The main simulation step of the circuit has to be executed at most L times since the simulated circuit has a depth of at most L.
> > >
> > > The transformer family is L-uniform: Its only aspect that depends on N is the size of the matrices, which needs to grow with the PEs.
> > > The counters required to construct such matrices can be implemented in log-space.
> > >
> > > **Note**: Lem. C.3 also very concisely *describes the proof of our Lem. 4.1 as well*: The construction from [2] is equivalent to our Lem. C.3, as it simulates the circuit directly encoded in the input PEs exactly like our Lem. C.3.
> > >
> > > [1] Svete, A. and Sabharwal, A. On the reasoning abilities of masked diffusion language models. https://arxiv.org/abs/2510.13117
> > >
> > > [2] London, C. and Kanade, V. Pause tokens strictly increase the expressivity of constant-depth transformers. https://openreview.net/forum?id=eG5oh8l1WZ.

---

### Decision · Program_Chairs · 2026-04-30

**Decision:**

Accept (regular)

**Comment:**

The paper is somewhat narrow in scope. I am also somewhat on the fence about expressivity results in general, as I don't find them technically challenging anymore. However, all reviewers recommended at least a weak accept, so given the novelty, even if it is limited in scope, I recommend weak acceptance.